

# Constraining Aerosol-Cloud Adjustments by Uniting Surface Observations with a Perturbed Parameter Ensemble

August Mikkelsen[1], Daniel T. McCoy[1], Trude Eidhammer[2], Andrew Gettelman[3], Ci Song[1], Hamish Gordon[4], Isabel L. McCoy[5, 6]

[1]Department of Atmospheric Science, University of Wyoming, Laramie, WY, USA
[2]NSF National Center for Atmospheric Research, Boulder, CO, USA
[3]Pacific Northwest National Laboratory, Richland, WA, USA
[4]Department of Chemical Engineering and Center for Atmospheric Particle Studies, Carnegie Mellon University, Pittsburgh, PA, USA
[5]Cooperative Institute for Research in Environmental Sciences, University of Colorado, Boulder, CO, USA
[6]NOAA Chemical Sciences Laboratory, Boulder, CO, USA

*Correspondence to*: August Mikkelsen (amikkels@uwyo.edu)

**Abstract.** Aerosol-cloud interactions (aci) are the largest source of uncertainty in inferring the magnitude of future warming consistent with the observational record. The effective radiative forcing due to aci (ERFaci) is dominated by liquid clouds and is composed of two terms: the change in cloud albedo due to redistributing liquid over a larger number of cloud droplets ($N_d$) and the change in cloud macrophysical properties due to changes in cloud microphysics. These terms are respectively referred to as the radiative forcing due to aci (RFaci) and aerosol-cloud adjustments. While the magnitude of RFaci is uncertain, its sign is confidently negative and results in a cooling in the historical record. In contrast, the adjustment of cloud liquid water path (LWP) to enhanced $N_d$ and associated radiative forcing is uncertain in sign. Increased LWP in response to increased $N_d$ is consistent with precipitation suppression while decreased LWP in response to increased $N_d$ is consistent with enhanced evaporation from cloud top. Observational constraints of these processes are poor in part because of causal ambiguity in the relationship between $N_d$ and LWP. To better understand this relationship, precipitation (P), $N_d$, and LWP surface observations from the Eastern North Atlantic (ENA) atmospheric observatory are combined with the output from a perturbed parameter ensemble (PPE) hosted in the Community Atmosphere Model version 6 (CAM6). This allows causal interpretation of observed covariability. Observations of precipitation and cloud from ENA constrain the range of possible LWP aerosol-cloud adjustments relative to the prior from the PPE by 15%, resulting in a global value that is confidently positive (a historical cooling) ranging from 2.1 to 6.9 g/m$^2$. It is found that observed covariability between $N_d$ and LWP is dominated by coalescence scavenging and that this observed covariability is not strongly related to aerosol-cloud adjustments.

## 1 Introduction

Atmospheric aerosols affect the global radiation budget through direct interactions with radiation and indirect interactions via clouds. Aerosol-cloud interactions (aci) are facilitated by aerosols serving as surfaces for water vapor to





condense onto, forming cloud droplets. These aerosols are called cloud condensation nuclei (CCN) and are essential for forming clouds in the troposphere (Gordon et al., 2023; Mason, 1960; Wilson, 1900).

While many CCN have natural sources, such as dust and sea spray (e.g., Carslaw et al., 2013), there are also CCN emitted from anthropogenic activities, including increased emission of carbonaceous aerosols (Hamilton et al., 2018) and sulfur dioxide (Charlson et al., 1992). Changing the amount of CCN in a cloud can change the droplet number concentration ($N_d$) of the cloud, shifting the cloud's albedo (Twomey, 1974). This is referred to as the radiative forcing from aci (RFaci, following notation from Bellouin et al., 2020). By affecting cloud and precipitation processes, changes in $N_d$ driven by CCN

can also change macrophysical cloud properties such as cloud liquid water path (LWP) (Ackerman et al., 2004; Albrecht, 1989). Changes in cloud macrophysics driven by changes in cloud microphysics in response to anthropogenic aerosols are referred to as aerosol-cloud adjustments. The sum of RFaci and forcing due to aerosol-cloud adjustments is termed the effective radiative forcing due to aci (ERFaci).

    Overall, there is high confidence that anthropogenic aerosols led to cooling during the historical record (Bellouin et

al., 2020). Aerosol cooling since the Industrial Revolution has offset warming from anthropogenic greenhouse gas emissions (Andreae et al., 2005; Charlson et al., 1992), but the degree to which warming has been offset is uncertain. Because of this gap in our knowledge, it is difficult to know the true sensitivity of Earth's surface temperature to greenhouse gas emissions (Forster, 2016; Watson-Parris and Smith, 2022). Aerosol cooling is dominated by ERFaci (Bellouin et al., 2020). Uncertainty in RFaci and aerosol-cloud adjustments both contribute to uncertainty in ERFaci. Uncertainty in the radiative forcing due to aerosol-

cloud adjustments based on observations and global modelling outpaces uncertainty driven by RFaci (Bellouin et al., 2020; Gryspeerdt et al., 2020; Heyn et al., 2017). The range of predicted future climate consistent with the historical record motivates developing constraints on ERFaci and in particular the sign and amplitude of the large aerosol-cloud adjustments forcing term (Andreae et al., 2005; Watson-Parris and Smith, 2022).

    There are several factors that contribute to the uncertainty of RFaci and aerosol-cloud adjustments. While the basic

understanding of what processes set aci and aerosol-cloud adjustments is good (Ackerman et al., 2004; Albrecht, 1989; Bretherton et al., 2007; Khairoutdinov and Kogan, 2000; Mülmenstädt and Feingold, 2018; Wood, 2012), these processes operate at small spatial and temporal scales that cannot be resolved by the global models that we rely on to calculate forcing. This scale mismatch means that we must parameterize these processes in global climate models (GCMs). This results in parametric uncertainty related to how a given process is parameterized (Regayre et al., 2018). It also results in structural

uncertainty related to which processes are parameterized and represented in a given GCM (Regayre et al., 2023). In addition to uncertainty related to how microscale aerosol, cloud, and precipitation processes translate to the global scale, our ability to constrain ERFaci is hindered by our lack of knowledge regarding the pre-industrial (PI) baseline. We lack observations of PI aerosol spatial distribution, emission, and composition outside of a few regions (Hamilton et al., 2014) that maintain a pristine state in the present day (PD). This lack of observational constraint of the baseline PI atmosphere drives substantial uncertainty

in forcing due to aci (Carslaw et al., 2013; McCoy et al., 2020b).



To narrow uncertainty in ERFaci we need to confront the GCMs that we rely on for calculations of ERFaci with observations of aerosol, clouds, and precipitation to identify whether there are parameter combinations that agree with observations and if our GCMs are structurally deficient (Ghan et al., 2016; Mülmenstädt and Feingold, 2018). A broad issue is that the causality linking aerosol, clouds, and precipitation is complex (Fons et al., 2023; Gryspeerdt et al., 2019; McCoy et al., 2020a; Stevens and Feingold, 2009). We provide a schematic illustration of some of the causal linkages in this aerosol-cloud-precipitation system in Figure 1. Coalescence scavenging of aerosol and cloud droplets further confounds the relationship linking cloud droplet number to liquid cloud properties (i.e., through aerosol-cloud adjustments that can increase and decrease LWP). Except in very specific situations (Christensen et al., 2022), we cannot untangle this causality using observations alone. In this study we unite a model where causality can be explicitly determined with observations of clouds and precipitation where we must infer causality.

We examine the adjustment of cloud LWP to changes in $N_d$. When evaluating how increased aerosol affects cloud liquid water content via $N_d$, there are two main effects of changes on $N_d$ theorized to play a substantial role in setting ERFaci (Mülmenstädt and Feingold, 2018). The first is precipitation suppression, wherein the decrease in average droplet size from increased $N_d$ reduces the precipitation production of a cloud, increasing the cloud's LWP (Albrecht, 1989). The second process is size-dependent evaporation and entrainment, wherein the increased $N_d$ may increase entrainment or evaporation at the cloud-top, reducing cloud liquid water content (Bretherton et al., 2007; Hill et al., 2009; Wang and Albrecht, 1994; Wang et al., 2003; Xue and Feingold, 2006).

Previous observations characterizing the sensitivity of LWP to $N_d$ have shown a positive correlation between $N_d$ and LWP in low-$N_d$ clouds and a negative correlation in high-$N_d$ clouds, with an average negative sign in oceanic cloud (Fons et al., 2023; Glassmeier et al., 2021; Gryspeerdt et al., 2019). However, it is challenging to understand cloud susceptibility to $N_d$ based on observations without accounting for precipitation. Coalescence scavenging (via precipitation) is a strong controller on $N_d$ in marine low clouds (Kang et al., 2022; Wood et al., 2012) and CCN below the cloud may be removed by wet scavenging of aerosol (Textor et al., 2006). These relationships and the observed correlations between clouds and precipitation from observations from the East North Atlantic (ENA) (Wood et al., 2015) atmospheric observatory, our surface observations source, are shown for reference (Figure 1).



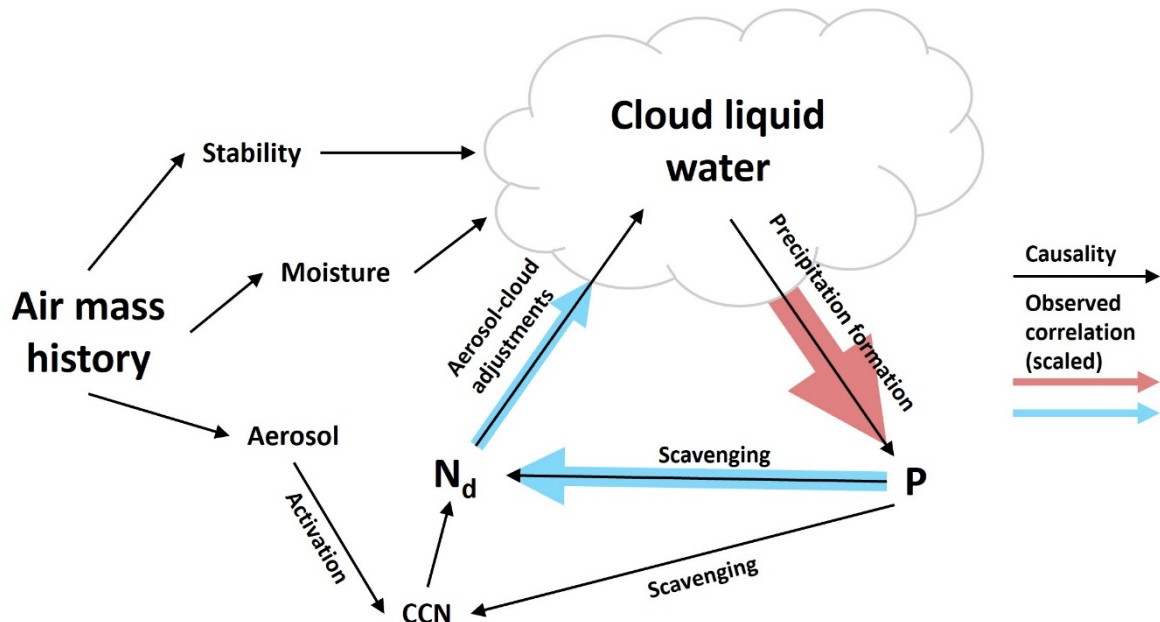

**Figure 1** *A schematic describing the causal links on aerosol-cloud adjustments. The blue or red shading indicates a positive or negative correlation between the two variables at ENA and the size of the shaded arrow indicates the relative magnitude of the correlation at ENA. This data is further detailed in Figure 7.*

While there have been many studies inferring cloud adjustments from LWP sensitivity to $N_d$ observed by satellites (Amiri-Farahani et al., 2017; Christensen et al., 2017; Fons et al., 2023; Gryspeerdt et al., 2017, 2019; Lebsock et al., 2008; McCoy et al., 2020a), there are comparatively fewer studies of aerosol-adjustments from a surface perspective (Chiu et al., 2021; Feingold et al., 2003; Gettelman et al., 2020; McComiskey and Feingold, 2012; Wu et al., 2020). There are benefits and drawbacks to the use of surface observations. An obvious drawback to using surface observations is that they only provide a limited sampling of the atmosphere relative to a satellite. Another drawback related to sampling is that it is unclear how surface measurements scale to a GCM grid cell (McComiskey et al., 2009; Mülmenstädt and Feingold, 2018). However, surface observations have several benefits in terms of observing clouds and precipitation. Commonly-used satellite $N_d$ products have sparse airborne validation and potentially large systematic uncertainties in some cloud regimes (Ahn et al., 2018; Grosvenor et al., 2018; Gryspeerdt et al., 2022; Kang et al., 2021; McCoy et al., 2018). One source of uncertainty in satellite $N_d$ is due to lack of homogeneity in the satellite footprint (Grosvenor and Wood, 2014). Surface remote-sensing has a substantially smaller footprint (McComiskey et al., 2009), which reduces the uncertainty inherent in passive retrieval-based calculations of $N_d$ (Cho et al., 2015; Grosvenor et al., 2018; Nakajima and King, 1990). Precipitation is challenging to observe from space (Kidd and Huffman, 2011; Pradhan et al., 2022; Sun et al., 2018). The ability to directly observe precipitation flux at the surface is uniquely advantageous. There are also benefits in surface observations of LWP – the retrieval used in this work (detailed in



Section 2.1.1) utilizes an ensemble of instruments to observe LWP, allowing for higher confidence than a large-footprint passive microwave radiometer. In concert with the surface P and $N_d$ retrievals, we have a suite of fine-resolution observations
for analysis.

Here, we constrain aerosol-cloud adjustments based on observable properties sampled at ENA: cloud and precipitation state variables and their covariances. A perturbed parameter ensemble (PPE) hosted in a GCM is used to define the causal inference from observations. Surface observations are used to provide a constraint on global-mean aerosol-cloud adjustments in LWP. Section 2 describes observational data and the PPE used. Section 3.1 describes the framework used in
this study to provide causal inference from observed cloud and precipitation. Section 3.2 constrains the PPE using observations. Section 3.3 provides a constraint on global-mean aerosol-cloud adjustments. Section 4 discusses the results and provides suggestions for future studies. Section 5 summarizes the conclusions.

## 2 Data and Methods

### 2.1 Observations

We leverage surface remote sensing and in situ observations from the Atmospheric Radiation Measurement (ARM) Eastern North Atlantic (ENA) observatory (Wood et al., 2015). ENA is located in the northeastern Atlantic Ocean approximately 1,000 miles (~1,600 km) west of Portugal on Graciosa Island in the Açores.

Use of surface-based observations and the selection of ENA is motivated by logistical and scientific concerns. Surface observations provide a unique set of strengths that align with the framework for constraining aerosol-cloud adjustment strength
as described above. Our underlying constraint framework is not dependent on the source of observations of LWP, P, and $N_d$. However, we argue that surface observations are better suited for this problem than spaceborne remote sensing despite the much larger data volume and coverage afforded by spaceborne remote sensing. Precipitation rates are challenging to measure from space, especially for lighter precipitation rates (Pradhan et al., 2022). Surface measurements of precipitation rate, in contrast, are a direct measurement, although they still struggle with observing very light precipitation and cannot observe virga.
Liquid cloud microphysical state can be measured remotely from space and the surface. The calculation of $N_d$ from remote sensing is based on the assumption of a homogeneous cloud within the sensor footprint (Grosvenor et al., 2018; Nakajima and King, 1990). Aircraft observations of $N_d$ are in reasonable agreement with spaceborne estimates in homogeneous cloud, but the agreement degrades in more heterogeneous cloud (Gryspeerdt et al., 2022). The sensor footprint of surface-based remote sensing of $N_d$ is drastically smaller and aircraft evaluation suggests minimal impacts from changes in cloud heterogeneity
(Zhang et al., 2023).

ENA is one of three surface sites administered by ARM where observations of LWP, $N_d$, and P are available. The other locations are ARM Southern Great Plains site (SGP) centered near Lamont, Oklahoma and the Layered Atlantic Smoke Interactions with Clouds (LASIC) field campaign that took place on Ascension Island in the central Atlantic. Maritime liquid clouds are a significant contributor to the uncertainty surrounding $ERF_{aci}$ (Bellouin et al., 2020; Carslaw et al., 2013; McCoy





et al., 2017; Wall et al., 2022, 2023) and we argue that a marine environment provides more information about the cloud and precipitation processes driving global aerosol-cloud adjustments. This suggests that SGP is less relevant to our current analysis. Between ENA and LASIC, ENA has a significantly larger pool of observations due to its considerably longer observational period, with LASIC only providing two years of data compared to ENA's nine (at the time of writing). Further, LASIC observes layers of carbonaceous aerosols in the free troposphere from Southern Africa, the largest biomass burning region in the world

(Zuidema et al., 2015). This unusual atmospheric aerosol regime adds the complexity of substantial aerosol semi-direct effects along with aerosol-cloud adjustments and may not be representative of a broader global regime. Bearing these points in mind, we see ENA as the most suitable observatory for the purposes of this study.

While surface observations provide direct measurements of precipitation fluxes and are essentially looking through a much shorter pathlength in the atmosphere to remotely sensed cloud properties, due to their nature, their sampling is limited

in extent compared to spaceborne remote sensing. For developing a constraint, understanding the systematic uncertainty from observations is much more important than understanding the random uncertainty; while it is easier to estimate an instrument's random uncertainty (e.g. by having two instruments measure the same thing), because it scales with $\frac{1}{\sqrt{N}}$ where $N$ is the number of observational samples, our random uncertainty goes towards zero quickly over the years of data recorded. Structural uncertainty is a lot harder to estimate for these observations. Unfortunately, no published systematic uncertainties could be

found for the observational products used. Though the observatory has been recording data since it was established in May 2014, we are limited to observations from October 2014 through October 2019 due to the combined availability of the three datasets detailed below. In all cases, data is averaged from its original time resolution to a 3-hour resolution. In this study we examine the cloud and precipitation properties highlighted in Figure 1: LWP, P, and $N_d$. We briefly describe the observational data sets used to quantify each property below.


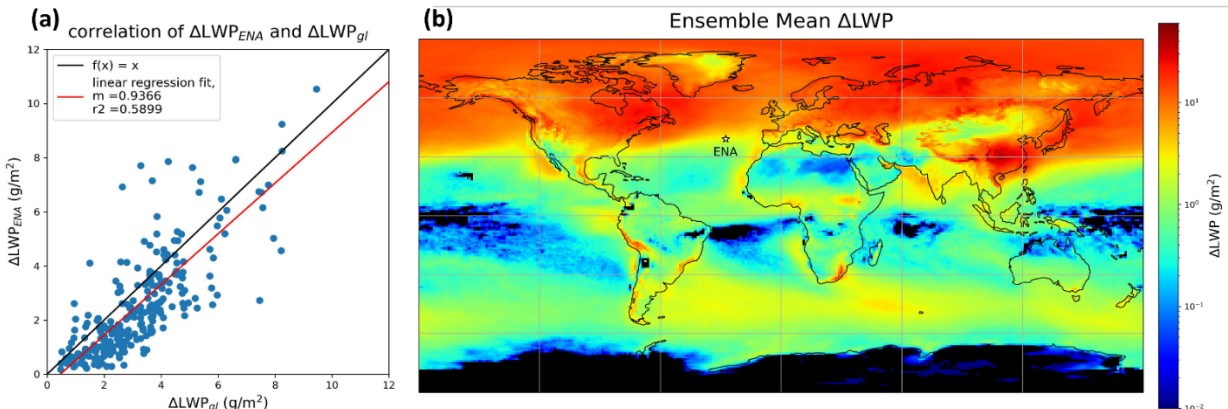

**Figure 2** (a) *ENA ΔLWP regressed on global ΔLWP with a slope of 0.94 and an $R^2$ of 0.60. (b) a map of PPE mean ΔLWP, with ENA's location marked with a star.*





### 2.1.1 Liquid water path

Cloud macrophysical state is characterized by LWP. Observations of LWP are provided by the Microwave Radiometer Retrievals with MWRRET Version 2 (MWRRETv2) value-added product (VAP) at ENA. In the MWRRETv2 VAP, LWP is retrieved at a ~15 second resolution with a physical-iterative algorithm detailed in Turner et al., 2007 that utilizes microwave brightness temperatures from the on-site 3-channel microwave radiometer and radiosonde temperature, pressure, and humidity profiles (launched three times daily and interpolated to 1-minute temporal resolution).

### 2.1.2 Precipitation

The ARM Video Disdrometer Quantities VAP VDISQUANTS (Hardin et al., 2020) provides observations of surface rain rate. Surface rain flux is observed at 1-minute intervals. While the video disdrometer is considered to be reliable and is frequently used as the truth for validation of satellite retrievals of rainfall (Raupach and Berne, 2015; Schuur et al., 2001; Tokay et al., 2020), the instrument may miss small drops, including those within the drizzle domain, due to wind-induced error (Nešpor et al., 2000). The surface precipitation measurements are inherently limited in that they miss virga because the precipitation evaporates before reaching the surface. Given drizzle and virga's prevalence in ENA's climatology (Wu et al., 2020), this may constitute a component of the sink of cloud water through precipitation. Supplementing surface flux observations of precipitation with radar would provide an estimate of the virga and drizzle sink term, but also require the implementation of a radar simulator (Silber et al., 2022), which is beyond the scope of our current study.

### 2.1.3 Droplet number concentration

Retrievals of $N_d$ from the ARM Droplet Number Concentration VAP NDROP are calculated following the method described in McComiskey et al. (2009). This method uses cloud optical depth obtained from a multifilter rotating shadowband radiometer (MFRSR); cloud base temperature and pressure from interpolated radiosonde observations; LWP from the microwave radiometer; and cloud boundary information from the Active Remote Sensing of Clouds (ARSCL) VAP. Because the MFRSR requires sunlight for its retrieval, $N_d$ retrievals are only available during the daytime. The $N_d$ calculated from NDROP compares favorably with aircraft and other surface remote sensing $N_d$ retrievals, but it tends to overestimate $N_d$ in broken cloud and low LWP regimes (Zhang et al., 2023).

### 2.2 The Sixth Community Atmosphere Model (CAM6) Perturbed Parameter Ensemble (PPE)

In GCMs, processes that take place on smaller scales than the model grid size (typically ~100 km) must be parameterized. Parameterizations are a source of uncertainty because (i) the uncertainty in the coefficients in the parameterization (parametric uncertainty) and (ii) the uncertainty in how processes are represented mathematically within the model and which processes are represented (structural uncertainty). While structural uncertainty is difficult to quantify (Regayre et al., 2020, 2023), we can use perturbed parameter ensembles (PPEs) to systematically explore parametric uncertainties in GCMs (Lee et al., 2011; Sexton et al., 2021).

We leverage a PPE hosted in Community Earth System Model version 2's (CESM2) atmospheric component, the sixth Community Atmosphere Model (CAM6) (Duffy et al., 2023; Eidhammer et al., 2024; Song et al., 2024) . The CAM6 PPE is





utilized as a tool to link the strength of the LWP adjustment between the PI and the PD atmosphere to present day variability

in clouds and precipitation.

Following the setup in Eidhammer et al., 2024, the CAM6 PPE varies 45 parameters that are sampled across 263 ensemble members. The parameters come from 4 physics schemes: Cloud Layers Unified By Binormals (CLUBB; Golaz et al., 2002), version 2 of the Morrison and Gettelman (2008) scheme (MG2; Gettelman & Morrison, 2015), the Modal Aerosol Model ("Aerosol" in Table 1; Liu et al., 2012), and the Zhang-McFarlane deep convection scheme (ZM; G. J. Zhang &

McFarlane, 1995). Each ensemble member in the PPE has a different, random, combination of parameter settings. These parameter combinations are generated using Latin hypercube sampling in order to efficiently fill the uncertainty space between parameters (Eidhammer et al., 2024; Lee et al., 2011). The varied parameters, along with their default values (i.e. the values in CAM6) and bounds, are detailed in Table 1.



| Physics Scheme | Parameter Name | Description | Default | Min | Max | Units |
|---|---|---|---|---|---|---|
| CLUBB | clubb_C2rt | Damping on scalar variances | 1.0 | 0.2 | 2 | - |
| | clubb_C6rt | Low skewness in C6rt skewness function | 4.0 | 2.0 | 6 | - |
| | clubb_C6rtb | High skewness in C6rt skewness function | 6.0 | 2.0 | 8 | - |
| | clubb_C6thl | Low skewness in C6thl skewness function | 4.0 | 2.0 | 6 | - |
| | clubb_C6thlb | High skewness in C6thl skewness function | 6.0 | 2.0 | 8 | - |
| | clubb_C8 | Coef. #1 in C8 skewness Equation | 4.2 | 1.0 | 5 | - |
| | clubb_beta | Set plume widths for theta_l and rt | 2.4 | 1.6 | 2.5 | - |
| | clubb_c1 | Low Skewness in C1 Skw. | 1.0 | 0.4 | 3 | - |
| | clubb_c11 | Low Skewness in C11 Skw | 0.7 | 0.2 | 0.8 | - |
| | clubb_c14 | Constant for $u'^2$ and $v'^2$ terms | 2.2 | 0.4 | 3 | - |
| | clubb_c_K10 | Momentum coefficient of Kh_zm | 0.5 | 0.2 | 0.6 | - |
| | clubb_gamma_coef | Low Skw.: gamma coef. Skw | 0.308 | 0.25 | 0.35 | - |
| | clubb_wpxp_L_thresh | Lscale threshold, damp C6 and C7 | 60 | 20 | 200 | m |
| MG2 | micro_mg_accre_enhan_fact | Accretion enhancing factor | 1.0 | 0.1 | 10.0 | - |
| | micro_mg_autocon_fact | Autoconversion factor | 0.01 | 0.005 | 0.2 | - |
| | micro_mg_autocon_lwp_exp | KK2000 LWP exponent | 2.47 | 2.10 | 3.30 | - |
| | micro_mg_autocon_nd_exp | KK2000 autoconversion exponent | -1.1 | -0.8 | -2 | - |
| | micro_mg_berg_eff_factor | Bergeron efficiency factor | 1.0 | 0.1 | 1.0 | - |
| | micro_mg_dcs | Autoconversion size threshold ice-snow | 500e-06 | 50e-06 | 1.000e-06 | m |
| | micro_mg_effi_factor | Scale effective radius for optics calculation | 1.0 | 0.1 | 2.0 | - |
| | micro_mg_homog_size | Homogeneous freezing ice particle size | 25e-6 | 10e-6 | 200e-6 | m |
| | micro_mg_iaccr_factor | Scaling ice/snow accretion | 1.0 | 0.2 | 1.0 | - |
| | micro_mg_max_nicons | Maximum allowed ice number concentration | 100e6 | 1e5 | 10.000e6 | # kg$^{-1}$ |
| | micro_mg_vtrmi_factor | Ice fall speed scaling | 1.0 | 0.2 | 5.0 | m s$^{-1}$ |
| Aerosol | microp_aero_npccn_scale | Scale activated liquid number | 1 | 0.33 | 3 | - |
| | microp_aero_wsub_min | Min subgrid velocity for liq activation | 0.2 | 0 | 0.5 | m s$^{-1}$ |
| | microp_aero_wsub_scale | Subgrid velocity for liquid activation scaling | 1 | 0.1 | 5 | - |
| | microp_aero_wsubi_min | Min subgrid velocity for ice activation | 0.001 | 0 | 0.2 | m s$^{-1}$ |
| | microp_aero_wsubi_scale | Subgrid velocity for ice activation scaling | 1 | 0.1 | 5 | - |
| | dust_emis_fact | Dust emission scaling factor | 0.7 | 0.1 | 1.0 | - |
| | seasalt_emis_scale | Seasalt emission scaling factor | 10.0 | 0.5 | 2.5 | - |
| | sol_factb_interstitial | Below cloud scavenging of interstitial modal aerosols | 0.1 | 0.1 | 1 | - |
| | sol_factic_interstitial | In-cloud scavenging of interstitial modal aerosols | 0.4 | 0.1 | 1 | - |
| ZM | cldfrc_dp1 | Parameter for deep convection cloud fraction | 0.1 | 0.05 | 0.25 | - |
| | cldfrc_dp2 | Parameter for deep convection cloud fraction | 500 | 100 | 1.000 | - |
| | zmconv_c0_lnd | Convective autoconversion over land | 0.0075 | 0.002 | 0.1 | m$^{-1}$ |
| | zmconv_c0_ocn | Convective autoconversion over ocean | 0.03 | 0.02 | 0.1 | m$^{-1}$ |
| | zmconv_capelmt | Triggering threshold for ZM convection | 70 | 35 | 350 | J kg$^{-1}$ |
| | zmconv_dmpdz | Entrainment parameter | -1.0e-3 | -2.0e-3 | -2.0e-4 | m$^{-1}$ |
| | zmconv_ke | Convective evaporation efficiency | 1.0e-5 | 1.0e-6 | 1.0e-5 | (kg m$^{-2}$ s$^{-1}$)$^{0.5}$ s$^{-1}$ |
| | zmconv_ke_lnd | Convective evaporation efficiency over land | 3.0e-6 | 1.0e-6 | 1.0e-5 | (kg m$^{-2}$ s$^{-1}$)$^{0.5}$ s$^{-1}$ |
| | zmconv_momcd | Efficiency of pressure term in ZM downdraft CMT | 0.7 | 0 | 1 | - |
| | mconv_momcu | Efficiency of pressure term in ZM updraft CMT | 0.7 | 0 | 1 | - |
| | zmconv_num_cin | Allowed number of negative buoyancy crossings | 1 | 1 | 5 | - |
| | zmconv_tiedke_add | Convective parcel temperature perturbation | 0.5 | 0 | 2 | K |

**Table 1** *Perturbed parameters from the CAM6 PPE. Table from Eidhammer et al., 2024.*

The CAM6 PPE uses the default CAM6 spatial resolution of 1.25°×0.9375°. Two scenarios are integrated to calculate adjustment strength in the CAM6 PPE: present day (PD) and preindustrial (PI). The PD scenario has emissions set for the year 2000 and the PI scenario has emissions set for 1850. Otherwise, the scenarios are the same. Wind and temperature fields are nudged to Modern-Era Retrospective analysis for Research and Applications, Version 2 (MERRA2) reanalysis (Molod et al., 2015) with a 24-hour relaxation time to set the large-scale circulation to be the same between ensemble members and

observations following previous studies comparing CAM6 to observations (Gettelman et al., 2020; Song et al., 2024).

    Data output is cumbersome for PPEs due to their large number of ensemble members. Higher frequency outputs and three-dimensional outputs are provided for ENA, allowing for direct comparison from observations. The outputs analyzed from the ENA surface site are detailed in Table 2.





***Table 2*** *CAM6 PPE outputs at ENA analyzed in this study.*

| CAM6 history field | Units | Description |
| --- | --- | --- |
| AWNC | $m^{-3}$ | Average cloud water number concentration |
| CLOUD | fraction | Cloud fraction |
| TGCLDLWP | $kg/m^2$ | Total grid-box cloud liquid water path |
| PRECC | m/s | Convective precipitation rate |
| PRECL | m/s | Large-scale (stable) precipitation rate |

Precipitation rate is calculated by adding together the convective precipitation rate (PRECC) and the large-scale precipitation rate (PRECL). Cloud droplet number concentration is calculated by dividing vertically-resolved, grid-average

cloud water number concentration (AWNC) by liquid cloud fraction (CLOUD), giving in-cloud droplet number concentration. A vertically distributed $N_d$ calculation is obtained and then averaged through liquid clouds in the column. We believe this to be the best analogue for NDROP from CAM6, although it should be noted that NDROP data is constrained to single layer clouds, and there is not a way to do this in a GCM. Model LWP (TGCLDLWP) is directly comparable to microwave radiometer LWP.

This study seeks to provide observational constraints on aerosol-cloud adjustments based on observations from ENA. The outputs used to calculate aerosol-cloud adjustments between PI and PD are detailed in Table 3.

***Table 3*** *CAM6 PPE outputs from the global domain analyzed in this study.*

| Global model output | Units | Description |
| --- | --- | --- |
| ACTNL | $m^{-3}$ | Average Cloud Top droplet number |
| FCTL | fraction | Fractional occurrence of cloud top liquid |
| TGCLDLWP | $kg/m^2$ | Total grid-box cloud liquid water path |
| PRECC | m/s | Convective precipitation rate |
| PRECL | m/s | Large-scale (stable) precipitation rate |

Due to space constraints, the three-dimensional output saved at ENA are not available over the globe and cloud-top $N_d$

is used in the calculation of a global $N_d$. To calculate cloud-top $N_d$, average cloud top droplet number (ACTNL) is divided by fractional occurrence of cloud top liquid (FCTL).

**2.3 Gaussian Process Emulation**

PPEs are useful for exploring the parametric uncertainty, but it would be prohibitively computationally expensive to explore that uncertainty space systematically because the number of ensemble members needed to regularly sample *p*



dimensional parameter space with $n$ samples in each dimension is $n^p$ (Lee et al., 2011). To explore parameter space efficiently we leverage the Earth System Emulator (ESEm) package (Watson-Parris et al., 2021) to build Gaussian Process (GP) emulators. By generating a multivariate distribution via GP regression of ensemble output (for example, LWP) on input ensemble parameters, we can emulate the relationship between sampled parameters and outputs. This is advantageous, as this sampling of the 45-dimensional parameter space across 263 PPE members is, while an even sampling of the space, a collection

of discrete points rather than smooth surface, so emulation is critical to provide statistically meaningful results and understand linkages between processes and model behavior. This approach has been used in many other model-evaluation studies (McCoy et al., 2020b; Regayre et al., 2018, 2020, 2023; Song et al., 2024; Watson-Parris et al., 2020).

      To create an emulator, training samples and testing samples from the PPE members are randomly chosen. Of the ensemble members available, 15 are set aside as the testing sample and the remainder are used for training. To validate these emulators,

the testing portion of the dataset withheld from training is compared with the emulator prediction. GPs carry an estimate of their own prediction confidence. If the emulator is good, 95% of the test data set should overlap with the 95% confidence interval for each prediction (Lee et al., 2011).

      After the emulators are validated, 10 million emulated ensemble members (hereafter referred to as "emulates") are created randomly sampling the 45 input parameters within their individual minimum and maximum bounds (see Table 1). This gives

a smooth surface to examine the model's uncertainty space.

      Because we have no observational record of PI cloud properties, we use the difference between PD and PI PPE scenarios to make inferences about the PI to PD adjustment strength. When discussing the difference in a modeled quantity across PD and PI, $\Delta$ is used. E.g., $\Delta$LWP = PD LWP - PI LWP.

      We create emulators for ENA median ln LWP (med. ln LWP$_{ENA}$), ENA median ln N$_d$ (med. ln N$_{d, ENA}$), ENA mean-

state P ($\overline{P_{ENA}}$), ENA ($\frac{d \ln N_d}{d \ln P}$), ENA ($\frac{d \ln LWP}{d \ln P}$), ENA ($\frac{d \ln LWP}{d \ln N_d}$), the PD-PI change in average global LWP ($\Delta$LWP$_{gl}$), and the PD-PI change in average global N$_d$ ($\Delta$N$_{d, gl}$). We then observationally constrain the parameter range by removing each emulate that does not contain the observed value within the emulate's 95% confidence interval. In the case of the covariances, the observation and its 95% confidence interval from the standard error of the regression are used for constraint. For the other variables, because we are using average values from a large dataset (five years of continuous observations), we are unconcerned

with random uncertainty and only utilize the single observation value in constraint. For all testing emulates in all emulators, there is 100% overlap of PPE validation data with the emulates' 95% confidence interval (Figure 3). The linear regression fit and associated R$^2$ for each validation is also provided in Figure 3.





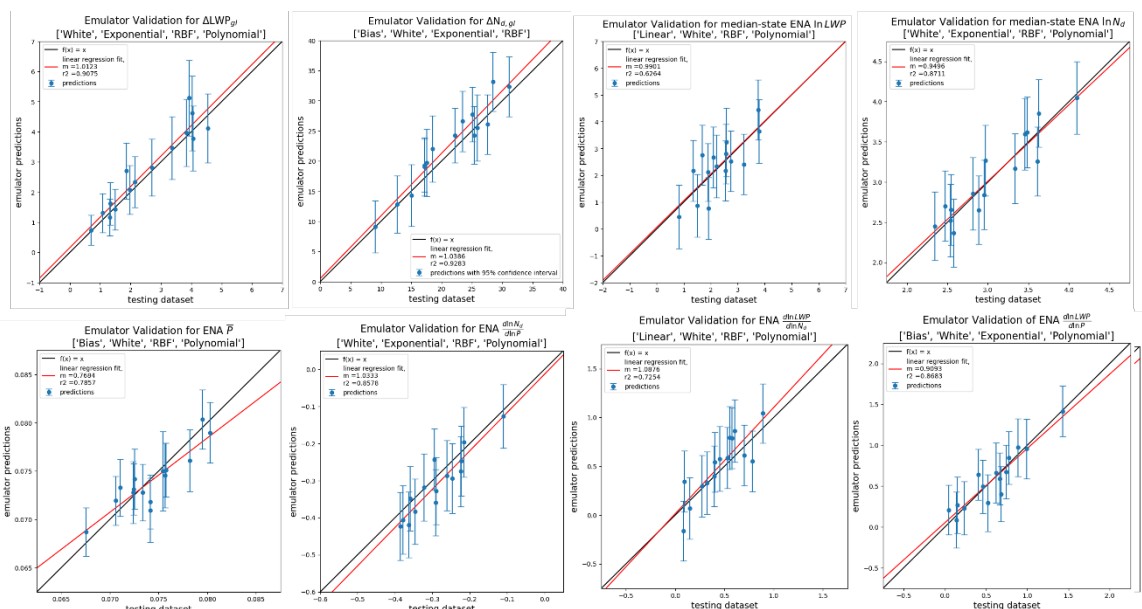

**Figure 3** *Validation plots for each emulator. For each emulator, the withheld test runs are plotted against emulator predictions. The blue vertical error bars are 95% confidence intervals of the emulate uncertainty. The explained variance and slope are noted for each emulator along with the kernels used to generate the emulator.*

# 3 Results

## 3.1 Developing a causally-aware framework for aerosol-cloud adjustments

The relationship between $N_d$ and LWP does not exist in isolation (Figure 1). Confounding sources of variability make it difficult to discern the causal link flowing from $N_d$ to LWP based on observed covariability between these terms. We illustrate this by examining the relationship between $N_d$ and LWP in CAM6 in the PI and PD in the northern hemisphere (NH) (Figure 4a). This is similar to previous studies examining the observed PD relationship between LWP and $N_d$ (Gryspeerdt et al., 2019), but within CAM6 we can contrast PD and PI relationships between LWP and $N_d$. The lack of agreement between the PI and PD illustrates that we cannot assume the observable covariation between $N_d$ and LWP is, on its own, predictive of the transient response of LWP to changes in $N_d$ driven by anthropogenic aerosol.

Given the numerous confounding factors acting on $N_d$ and LWP (Gryspeerdt et al., 2019; Stevens and Feingold, 2009), a more complex analysis than examining covariance between $N_d$ and LWP is required to isolate a causal relationship. Previous studies have described coalescence scavenging of droplets acting to create a negative correlation between LWP and $N_d$ (Gryspeerdt et al., 2019; McCoy et al., 2020a). To illustrate the importance of this confounding factor, we examine PI and PD LWP binned by precipitation rate (Figure 4b). In each bin of precipitation rate, coalescence scavenging is approximately constant. While holding coalescence scavenging constant, we contrast PI and PD LWP and PD LWP binned into the top and




bottom terciles of $N_d$ (Figure 4b). Precipitation suppression in CAM6 leads to higher LWP at higher $N_d$ and a constant rain
rate. For low rain rates, the high-$N_d$ regime has a distinctly higher LWP than its low-$N_d$ counterpart in the same precipitation
bin.

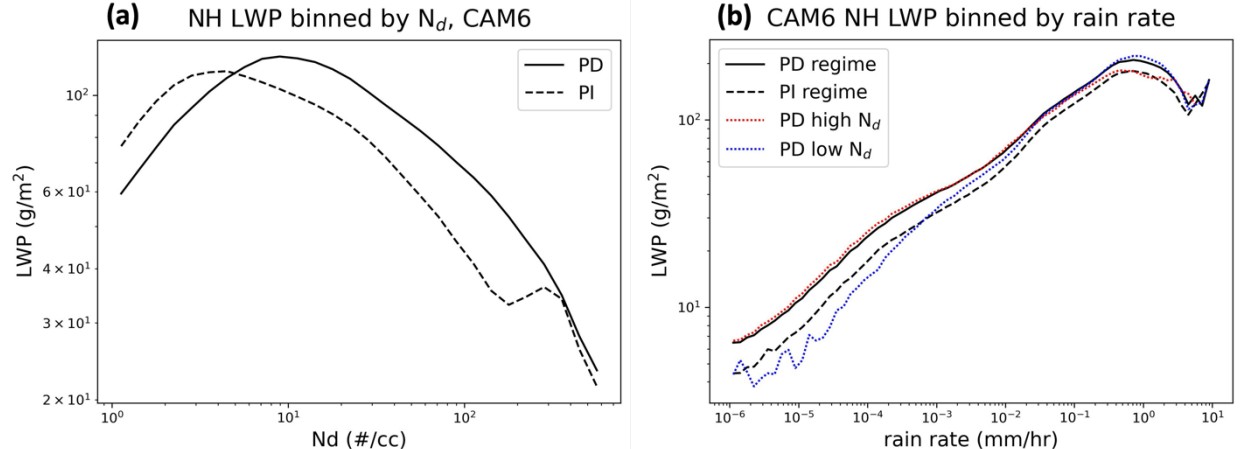

**Figure 4** *Northern hemisphere (30N-70N) PD and PI LWP in CAM6 binned by $N_d$ (a) and rain rate (b). In (b) PD LWP is shown separated into the top and bottom terciles of $N_d$. The northern hemisphere is used specifically in this figure to highlight the effect of aerosol-cloud adjustments, since this is the region that we would expect so see the most anthropogenic aerosol emissions (and thus the highest PD-PI difference).*

The relationship between LWP and $N_d$ in the PD is not predictive of PI to PD changes in LWP (Figure 4a). We cannot
rely on PD covariance between LWP and $N_d$ to predict aerosol-cloud adjustments and we need to consider non-causal sources
of covariance between LWP and $N_d$ in developing a constraint on aerosol-cloud adjustments from the PD (Mahfouz et al.,
2024; Mülmenstädt et al., 2024a, b). The following covariances, which are intended to contain information about processes
illustrated in Figure 1, are considered:

- $\frac{d \ln N_d}{d \ln P}$, for the below-cloud scavenging of droplets from precipitation.

- $\frac{d \ln LWP}{d \ln P}$, for autoconversion, the process by which cloud droplets collide with each other to form drizzle drops,
which ultimately leave the cloud via precipitation.

- $\frac{d \ln LWP}{d \ln N_d}$, for the observed susceptibility of cloud liquid water content to different droplet number concentrations.
This can be thought of as an "observed adjustments term", although as discussed above, it does not describe a
causal relationship between $N_d$ and LWP.

We argue, consistent with previous studies (Fons et al., 2023; Glassmeier et al., 2021; Gryspeerdt et al., 2019; McCoy
et al., 2020a; Mülmenstädt et al., 2024b), that to infer the strength of aerosol-cloud adjustments we need to consider the
confounding relationship that flows from LWP to precipitation and to $N_d$. By considering covariances between LWP and P
and $N_d$, we can estimate the strength of this term. To characterize aerosol-cloud adjustments in the context of these covariances





we need to have an underlying causal model. Here, we leverage the CAM6 PPE in this capacity to allow us to build a framework linking the aerosol-cloud adjustment due to anthropogenic aerosol to observed PD covariance between LWP, P, and $N_d$. We

constrain the PPE by the observed mean-states of LWP, P, and $N_d$ and the covariances between them (Table 4). By selecting the parameter space where PPE ensemble members agree with the quantities in Table 4 at ENA, we can link PD observations to the ΔLWP due to anthropogenic aerosol emissions.

*Table 4 Base-state variables and covariances at ENA used in this study to constrain aerosol-cloud adjustments.*

| Variable name | Description |
|---|---|
| median-state ln LWP | natural logarithm of the median-state liquid water path |
| median-state ln $N_d$ | natural logarithm of median-state droplet number concentration |
| mean-state P | the mean-state precipitation rate |
| $\dfrac{d \ln \text{LWP}}{d \ln \text{P}}$ | the covariance of the natural logarithm of liquid water path with the natural logarithm of precipitation rate |
| $\dfrac{d \ln \text{LWP}}{d \ln \text{N}_d}$ | the covariance of the natural logarithm of liquid water path with the natural logarithm of droplet number concentration |
| $\dfrac{d \ln \text{N}_d}{d \ln \text{P}}$ | the covariance of the natural logarithm of droplet number concentration with the natural logarithm of precipitation rate |

One concern is how relevant observations at ENA are to understanding global-mean aerosol-cloud adjustment and, by extension, ERFaci. However, across the CAM6 PPE cloud adjustments at ENA ($\Delta \text{LWP}_{\text{ENA}}$) are found to be correlated with global adjustments ($\Delta \text{LWP}_{\text{gl}}$) with a slope of 0.94 and an $R^2$ of 0.60 (Figure 2a). This correspondence between aerosol-cloud adjustments at ENA and global-mean aerosol-cloud adjustments is sensible because ENA straddles the border of the extratropics and subtropics (Figure 2b); we expect that the same aerosol, cloud, and precipitation processes being observed at

ENA are relevant over the other oceans in these regions where marine stratocumulus dominates.

In the PPE, we find differences in the predictive ability of $\Delta N_d$ for ΔLWP in the local (ENA) and global regimes. Specifically, we find that while $\Delta N_d$ alone is not a good predictor of ΔLWP in the global regime (as expected following **Figure 4**), it has more predictive ability in the ENA regime (Figure 5). This indicates that in CAM6 local-scale adjustments are sensitive to local perturbations in $N_d$, while global-scale adjustments are more influenced by physical processes.




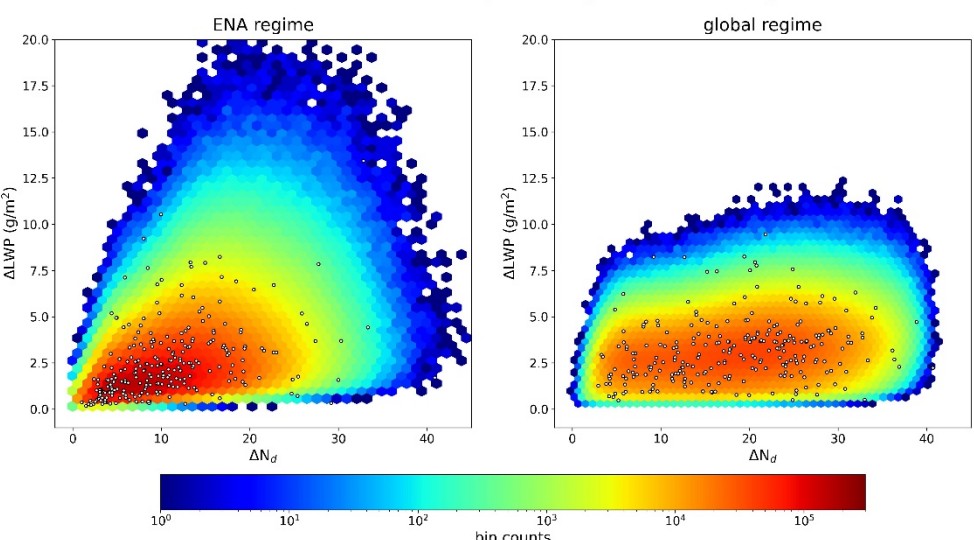

**Figure 5** *The distributions of emulated $\Delta N_d$ and $\Delta LWP$ in the global and ENA regimes are represented by the rainbow-colored hexbins. The original PPE ensemble members are represented by scattered white dots with black borders.*

### 3.2 Model-observation comparison

Before applying the framework described in the preceding section to constrain aerosol-cloud adjustments, we characterize PD $N_d$, LWP, and P at ENA in the observations and in the CAM6 PPE (Table 4). The observations are found to fall within the range of the PPE (Figure 6).

In the PPE and the observations, the mean-state of precipitation rate is used instead of the median-state because the video disdrometer cannot see extremely low precipitation rates, which are prevalent in the PPE data over ENA, consistent with most GCMs (Stephens et al., 2010). Differences between observed and PPE precipitation rate may also be due to sampling differences between averaged CAM6 data from a largely oceanic grid cell ~100 x 100 km² and observation data from a single point on an island.






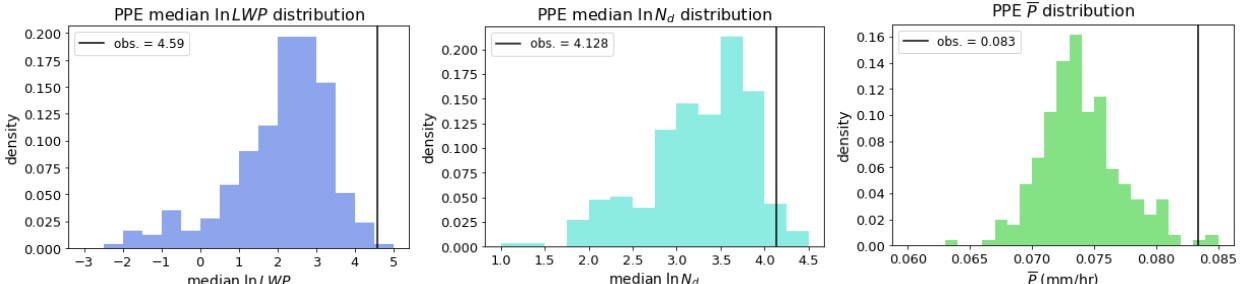

**Figure 6** *Distributions of median ln LWP, median ln $N_d$, and mean P across the PPE with the observational equivalents depicted with black bars.*

Covariances between variables are characterized by the linear regression slope of their constituent variables (e.g., $\frac{d \ln \text{LWP}}{d \ln \text{P}}$ is the slope of the regression of ln LWP on ln P). We observe $\frac{d \ln LWP}{d \ln N_d}$ to be -0.236 with a 95% confidence interval of +/- 0.051 (from the standard error of the linear regression), $\frac{d \ln \text{LWP}}{d \ln \text{P}}$ to be 0.338 with a 95% confidence interval of +/- 0.009,

and $\frac{d \ln N_d}{d \ln P}$ to be -0.258 +/- 0.092 (Figure 7). All values are unitless. These slopes were used to scale the shaded arrows in Figure 1. To match the 3-hour temporal resolution of the PPE data used to calculate the PPE covariances, we have binned the observations to 3 hours. As expected, there is a strong positive correlation between P and LWP with an r-value of 0.741 (Figure 7). Consistent with previous satellite-based studies there is a negative correlation between LWP and $N_d$ (Gryspeerdt et al., 2019). Consistent with our understanding of coalescence scavenging, there is a negative correlation between P and $N_d$ (Kang

et al., 2022; Wood et al., 2012). Observed $\frac{d \ln \text{LWP}}{d \ln \text{P}}$ and $\frac{d \ln N_d}{d \ln P}$ are closer to the PPE distribution means (Figure 8) while observed $\frac{d \ln \text{LWP}}{d \ln N_d}$ is on the very low end of PPE predictions, with an opposite sign compared to most of the PPE distribution. This result is discussed in more detail in Section 4.





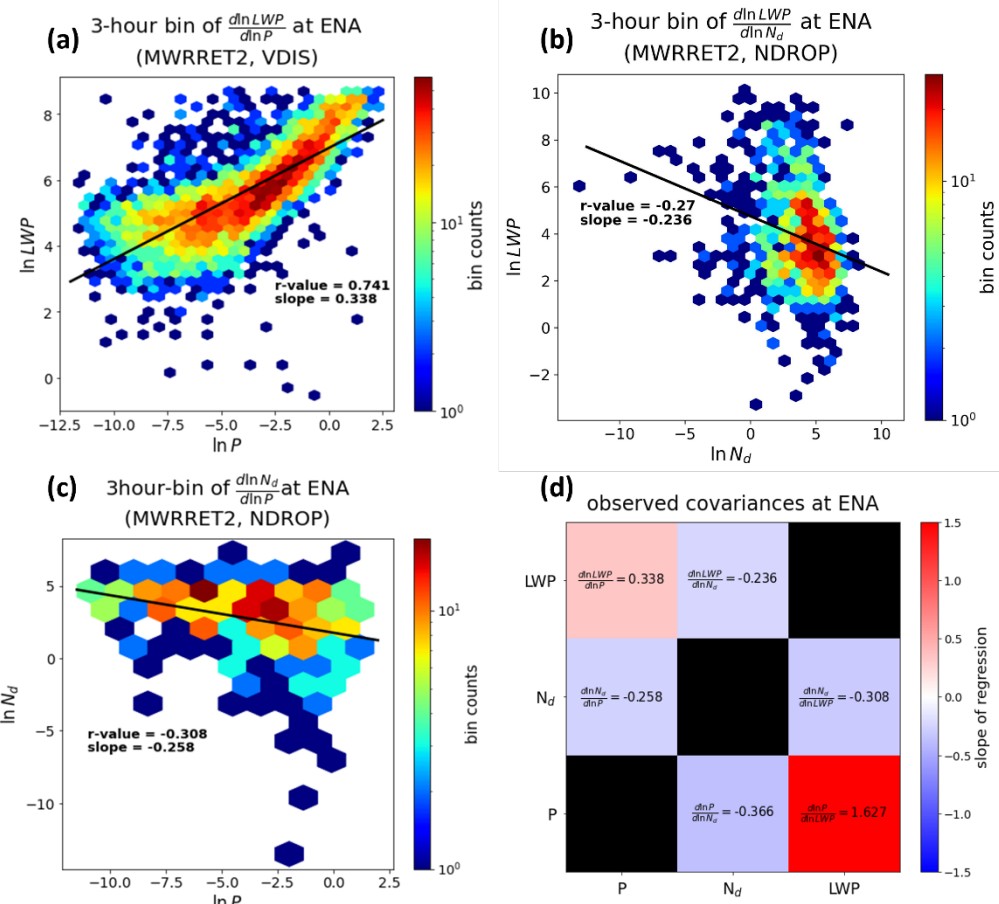

**Figure 7** *Observed covariances at ENA, derived with observations binned to 3-hour temporal resolution. 2D histograms relating LWP to P (a), LWP to $N_d$ (b), and $N_d$ to P (c). $\frac{d\ln LWP}{d\ln P}$, $\frac{d\ln LWP}{d\ln N_d}$, and $\frac{d\ln N_d}{d\ln P}$ derived from linear regressions are noted in a-c and summarized in (d).*

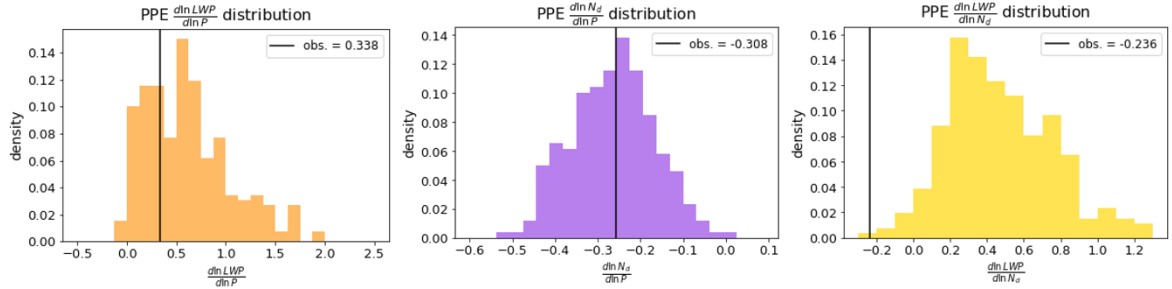

**Figure 8** *Distributions of $\frac{d\ln LWP}{d\ln P}$, $\frac{d\ln LWP}{d\ln N_d}$, and $\frac{d\ln N_d}{d\ln P}$ across the PPE with the observational equivalents depicted with black bars.*



We leverage the CAM6 PPE to understand linkages between covariances and states of LWP, $N_d$, and P and parameterized processes in CAM6. This is done by correlating values of perturbed parameters (Table 1) with mean-states and covariances (Table 4) across the PPE (Figure 9). While many of these correlations are low, there are stronger correlations associated with cloud and precipitation process parameters. This supports the utility of the framework in this study since it is picking out information about these processes.

We briefly discuss some of the stronger correlations between observables and processes and how these may link processes and observed quantities in a qualitative sense. Within CAM6, aerosol-cloud adjustments should occur through precipitation suppression operating through the autoconversion parameterization. This can be seen as a grouping of strong correlations related to 'micro_mg_autocon_' parameters (see Table 1 for descriptions). The inferred strength of coalescence scavenging ($\frac{d \ln N_d}{d \ln P}$) correlates strongly with the accretion enhancement factor (micro_mg_accre_enhan_fact). Mean-state $N_d$

is strongly correlated with the subgrid velocity and liquid activation parameters (micro_aero_npccn_scale, microp_aero_wsub_min, and microp_aero_wsub_cale). Several parameters are important for setting the median-state LWP and strong correlations can be seen relating median-state LWP to CLUBB parameters that relate to skewness in vertical velocity (clubb_c1, clubb_C8, and clubb_c14) which results in changes in cloud liquid content and, subsequently, reflectivity (Eidhammer et al., 2024; Guo et al., 2014). Mean state precipitation properties are correlated with several parameters in the

ZM convection scheme that are important for setting the amount and strength of convection and likely affect the creation of convective precipitation (zmconv_tiedke_add, zmconv_momcu, zmconv_ke_lnd, zmcomv_ke, zconv_dmpdz, and zmconv_capelmt).



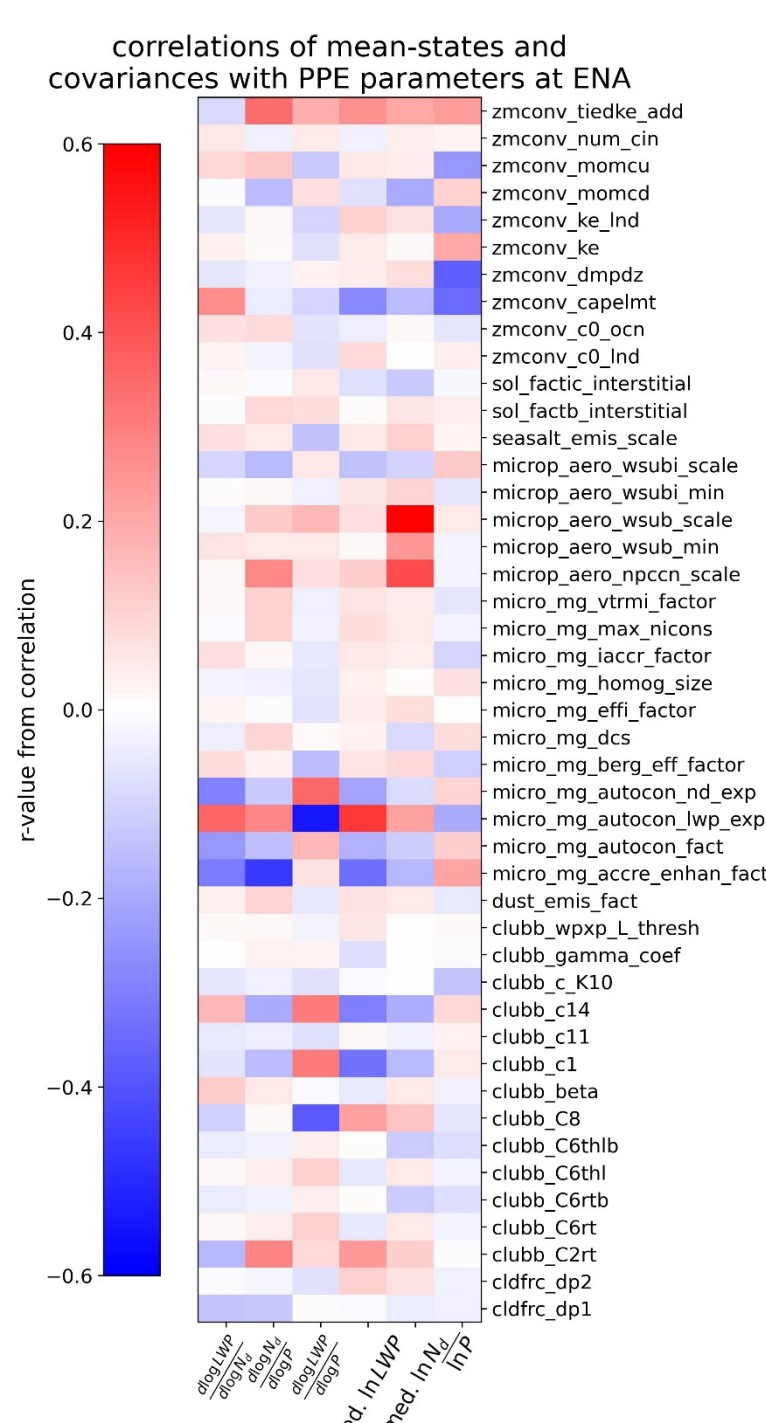

**Figure 9** *Correlations between covariances and PPE parameters at ENA. Note that the colorbar bounds span between -0.6 and +0.6, with the highest magnitude r-value recorded at 0.55.*



### 3.3 Constraining LWP adjustment strength from present-day observations

We seek to constrain aerosol-cloud adjustment strength in CAM6 by leveraging process-scale observations at ENA. As discussed in Section 3.1, we need to simultaneously consider relationships between P, LWP, and $N_d$. We constrain the PPE by the variables detailed in Table 4. Before constraining global adjustments, it is useful to understand how each observation is constraining its own variable within the emulated PPE. In Figure 10, distribution of each variable in the emulator field are shown overlaid by the distribution of emulates that are observationally constrained by that variable. Observations are found to

be within the PPE distribution for all variables.

When we use these same individual constraints to examine global-mean aerosol-cloud adjustments ($\Delta LWP_{gl}$), we see that observational constraints do not uniformly pull $\Delta LWP_{gl}$ one way or the other (Figure 11). The degree to which an observational constraint is effective at reducing the 95% confidence interval for $\Delta LWP$ is determined by (i) the distance between the observation and the mean of the distribution and (ii) the relative variance of the emulates within the distribution.

This illustrates why mean-state precipitation is such a powerful constraint: the observation is relatively far from the mean, out towards the right tail of the distribution (Figure 10e), while the average relative variance is relatively low. Intuitively, $\frac{d \ln LWP}{d \ln N_d}$ (Figure 10f) should be one of the strongest constraints on aerosol-cloud adjustments given its proximity to the processes responsible for aerosol-cloud adjustments (Figure 1) and its relatively large distance from the mean. However, this emulator was relatively uncertain (Figure 3) and the standard error from the linear regression was relatively high (Section 3.2), so the

observation remains within the permissible range for many emulates.

After discarding all invalid emulates, we are left with only the emulates that agree with the observations. This subset of emulates is the observationally-constrained dataset that is analyzed for most of the remainder of this paper.



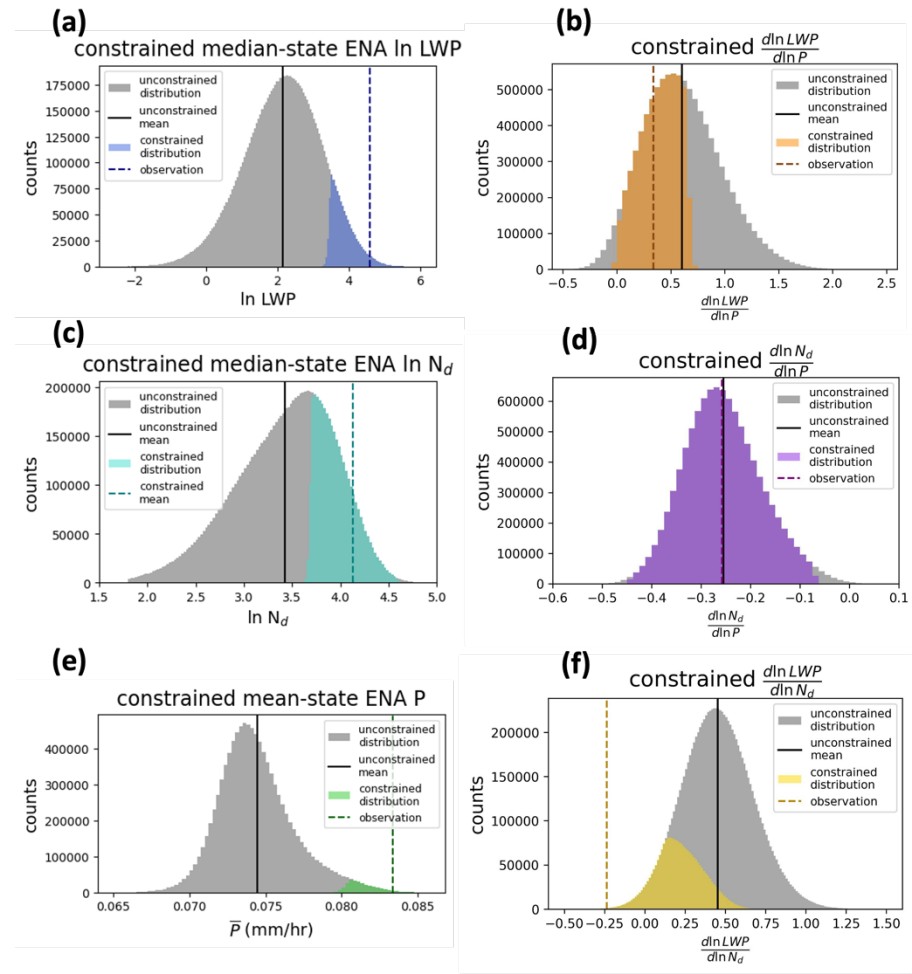

**Figure 10** *Distributions of PPE emulates with the observationally-constrained regions shaded in. For each plot, the variable is only being constrained by its associated observation. For instance, the shaded region in (a) is the subset of emulates that contain the observed value within their respective variances; in other words, the emulates of median-state ENA ln LWP that are observationally-constrained. (b), (c), (d), (e), and (f) are the same for their respective variables.*





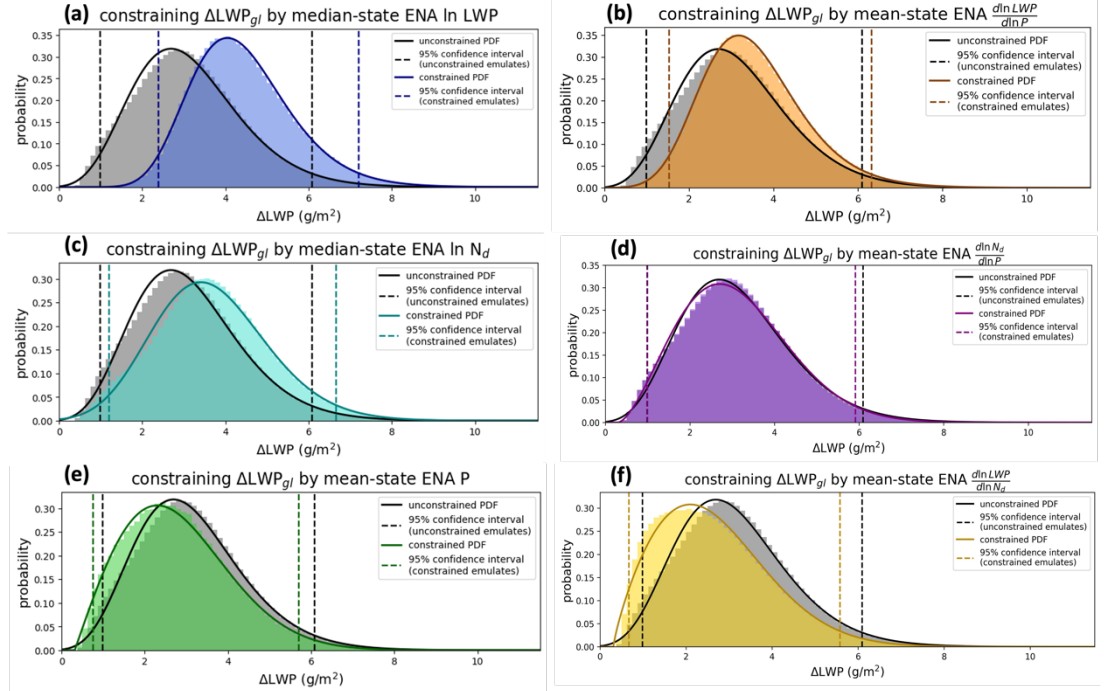

**Figure 11**, *$\Delta LWP_{gl}$ is constrained by individual constraints, with the color-shaded region in each plot representing the constrained distribution and the gray region representing the prior distribution. These constraints are (a) median-state ENA ln LWP, (b) $\frac{d \ln LWP}{d \ln P}$, (c) median-state ENA ln $N_d$, (d) $\frac{d \ln N_d}{d \ln P}$, (e) mean-state ENA P, and (f) $\frac{d \ln LWP}{d \ln N_d}$. Black vertical lines are the 95% CIs for unconstrained emulate distributions and colored vertical lines are the 95% CIs for constrained emulate distributions.*

Constraining $\Delta LWP$ by the variables in Table 4 removes the vast majority of emulates, leaving 11,053 (0.11%) of the original $10^7$ emulates. While this is a small fraction of the total number of prior emulates, 45 dimensions are being constrained and even moderate constraints in a few dimensions scale quickly. For instance, a fractional reduction in range of $f$ in $n$

dimensions scales as $f^n$ remaining emulates and the reduction described above is equivalent to constraining to 50% of the range of 6 parameters.

The constraints on the prior parameter ranges results in a constraint on $\Delta LWP_{gl}$ (Figure 12). The prior distributions of $\Delta LWP_{gl}$ ranges from 0.99 g/m² to 6.64 g/m² while the constrained $\Delta LWP_{gl}$ ranges from 2.08 g/m² to 6.87 g/m²; $\Delta LWP_{gl}$ is constrained by 15% (calculated by the change from confidence intervals). The observational constraint of $\Delta LWP_{gl}$ does not

strongly skew the distribution away from the CAM6 default adjustments.



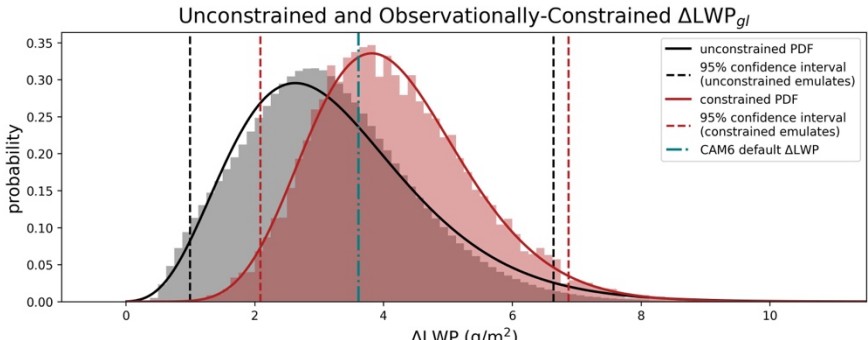

**Figure 12** *Constrained and unconstrained distributions for the global and ENA ΔLWP regimes. Unconstrained 95% confidence intervals are bounded by black dashed lines and constrained 95% confidence intervals bounded by red dashed lines.*

When examining the parameters constrained by the observations, we see substantial constraints in the distributions
of parameters in the CLUBB, MG2, and ZM physics schemes (Figure 13). This is consistent with the correlations between variables in Table 4 and the CAM6 parameters (Figure 9) as well as our *a priori* expectations based on underlying model physics. Autoconversion is the process through which precipitation is suppressed in aerosol cloud adjustments in CAM6 and we find that the associated terms within MG2 (micro_mg_autocon_fact, micro_mg_autocon_lwp_exp, micro_mg_autocon_nd_exp, and micro_mg_accre_enhan_fact; see Table 1 for details) are constrained in that the posterior
distribution is very different than the flat prior distribution for each parameter. Additionally, we find that several parameters that are important for setting the mean state of $N_d$ (micro_aero_npccn_scale, micro_aero_wsub_min, and micro_aero_wsub_scale); convective versus large-scale precipitation occurrence (cldfrc_dp2, zmconv_capelmt, zmconv_dmpdz, and zmconv_tiedke_add); and other boundary layer cloud properties (clubb_C2rt, clubb_c8, clubb_c14, clubb_c11, clubb_c1) (Eidhammer et al., 2024; Guo et al., 2014) are substantially constrained.




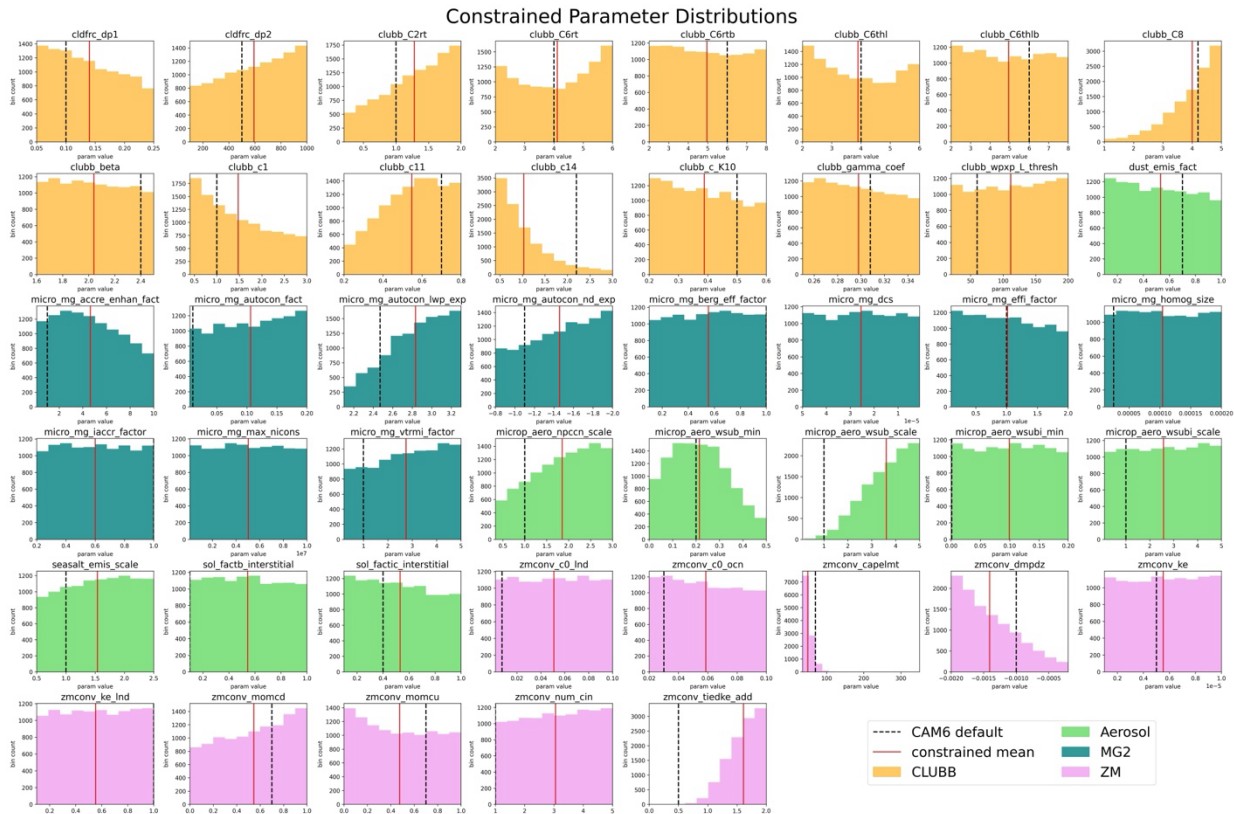

**Figure 13** *Grid of the distribution of PPE parameter values within the constrained set of 11,053 emulates, each distribution colored by the parameter category it is in as detailed in Section 2.2. These categories are CLUBB (orange), Aerosol (green), MG2 (teal), and ZM (pink). For more information on the parameters and their bounds, see Table 1.*

Finally, we investigate the relationship between the emulated distributions of $\Delta LWP_{gl}$ and $\frac{d \ln LWP}{d \ln N_d}$ (Figure 14) and whether their covariance, particularly the correspondence between negative $\frac{d \ln LWP}{d \ln N_d}$ and positive $\Delta LWP_{gl}$, is *causal*. $\Delta LWP_{gl}$ and $\frac{d \ln LWP}{d \ln N_d}$ are clearly related but the processes driving $N_d$ and LWP and their linkage via adjustments are more complex

than can be captured by a simple causal relationship between $N_d$ and LWP characterized by a linear regression of LWP on $N_d$. Figure 14Figure 14Figure 14

To interpret these results, it is useful to understand how autoconversion and accretion are parameterized in GCMs. In CAM6's MG2 (Gettelman and Morrison, 2015) and elsewhere (Jing et al., 2019; Michibata and Takemura, 2015), autoconversion is represented as a power law function of the form


$$R_{aut} = C_{aut} * L_c^{\alpha} N_d^{\beta} \qquad (1)$$



where $R_{aut}$ is the rate of autoconversion of droplets into rain; $L_c$ denotes cloud liquid water content; and $C_{aut}$, $\alpha$, and $\beta$ are constants. $C_{aut}$ is the *autoconversion enhancement factor*, represented in the model as *micro_mg_autocon_fact*; $\alpha$ alters the exponent on $L_c$, represented in the model as *micro_mg_lwp_exp*; and $\beta$ alters the exponent on $N_d$, represented in the model as

*micro_mg_nd_exp*. $\beta$ in the CAM6 PPE is a negative number with bounds between -2.0 and -0.8. $C_{aut}$ and $\alpha$ are positive in the CAM6 PPE with bounds of 0.005 to 0.2 and 2.10 to 3.30, respectively. Adjustments are driven by precipitation suppression as characterized by the exponent on $N_d$.

Accretion is parameterized in CAM6's MG2 (Gettelman and Morrison, 2015) and elsewhere (Michibata and Takemura, 2015) with the form


$$R_{accre} = C_{accre} L_c q_r \qquad (2)$$

where $C_{accre}$ is *micro_mg_accre_enhan_fact* and $q_r$ is the mixing ratio of drizzle. In the CAM6 PPE, *micro_mg_accre_enhan_fact* is a positive number with bounds of 0.0 and 10.0. Like autoconversion, accretion can be thought of as a sink of cloud water and scales negatively with $\Delta$LWP and $\frac{d \ln \text{LWP}}{d \ln N_d}$ (Figure 14e).

Observed covariability between $N_d$ and LWP is driven by coalescence scavenging and is strongly determined by the autoconversion enhancement factor and more moderately determined by the accretion enhancement factor. This is shown in Figure 14b-e, where it can be seen that *micro_mg_autocon_fact* scales primarily with $\frac{d \ln \text{LWP}}{d \ln N_d}$; *micro_mg_nd_exp* primarily with $\Delta$LWP; and *micro_mg_autocon_lwp* and *micro_mg_accre_enhan_fact* scale with both.

The relationship between $\Delta$LWP and autconversion parameters can be understood using the steady-state conceptual

model in Song et al. 2024 (their equation S1-5). In the PI and PD clouds are at a steady state balance between sources and sinks. The sink term is enforced by the large-scale moisture convergence, which is in turn enforced by the global pattern of sea surface temperature. Considering autoconversion to be the dominant sink term of cloud, the tendency from autoconversion should be approximately the same in PI and PD

$$C_{aut} * L_{c_{PD}}^{\alpha} N_{d_{PD}}^{\beta} = C_{aut} * L_{c_{PI}}^{\alpha} N_{d_{PI}}^{\beta} \qquad (3)$$


which can be solved for the change in $\ln L_c$

$$\Delta \ln L_c = -\frac{\beta}{\alpha} \Delta \ln N_d \qquad (4)$$





While highly idealized, this provides some insight into the behavior in Figure 14. The autoconversion scale factor ($C_{aut}$ or
*micro_mg_autocon_fact* in Figure 14c) does not impact the adjustment strength, but it does affect the covariance between N$_d$
and LWP through coalescence scavenging (Wood et al., 2012) by setting precipitation rate. This is consistent with the lack of
dependence of $\Delta$LWP on the autoconversion scale factor but the strong dependence of $\frac{d\ln \text{LWP}}{d\ln \text{N}_d}$ on this parameter in Figure 14c.
As expected, aerosol cloud adjustments scale very strongly with the N$_d$ exponent ($\beta$ or *micro_mg_nd_exp* in Figure 14d),
while the observed covariability between N$_d$ and LWP characterized by $\frac{d\ln \text{LWP}}{d\ln \text{N}_d}$ is not strongly affected by this term because
of its weak overall contribution to setting precipitation rates and, by extension, coalescence scavenging. This is consistent with
the strong dependence of $\Delta$LWP in Figure 14d on *micro_mg_nd_exp* and the lacking dependence of $\frac{d\ln \text{LWP}}{d\ln \text{N}_d}$ on this parameter.
The modulation of both adjustments and PD covariability between N$_d$ and LWP by $\alpha$ is less easily interpreted because both
the adjustment strength and PD N$_d$-LWP covariability are substantially affected by this term.

If we apply a similar logic to accretion, we get

$$C_{accre} * L_{c_{PD}} q_{r_{PD}} = C_{accre} * L_{c_{PI}} q_{r_{PI}} \qquad (5)$$

which can be solved for the change in $\ln L_c$

$$\Delta \ln L_c = -\Delta \ln q_r \qquad (6)$$

This can be read as the strength of adjustments from accretion being dependent on the change in precipitation rates, which we
have previously described to strongly scaled by autoconversion parameterizations. Although accretion is important for
understanding the change in adjustments, and indeed follows a similar behavior to *micro_mg_autocon_fact*, it is less easily
disentangled from the system than *micro_mg_autocon_fact* given its dependence on q$_r$, a variable modified by accretion's own
parameter *micro_mg_accre_enhan_fact* as well as the previously discussed autoconversion parameters. Following equations
(*2*) and (*6*, the negative correlation between *micro_mg_accre_enhan_fact* and both $\frac{d\ln \text{LWP}}{d\ln \text{N}_d}$ and $\Delta$LWP is expected: the rate of
accretion is an important part of setting precipitation rates (and by extension, the rate of coalescence), and modified
precipitation rates are the critical process for adjustments.

While precipitation suppression is the main control on adjustments in the CAM6 PPE, this does not project directly
onto $\frac{d\ln \text{LWP}}{d\ln \text{N}_d}$. The explained variance ($R^2$) in adjustment strength ($\Delta$LWP$_{gl}$) by $\frac{d\ln \text{LWP}}{d\ln \text{N}_d}$ (Figure 14a) is only 12%. This
highlights the importance of considering other confounding processes when attempting to use observed covariation between
N$_d$ and LWP as a constraint on aerosol cloud adjustments, as has been done in assessments of the total aerosol forcing (Bellouin
et al., 2020).



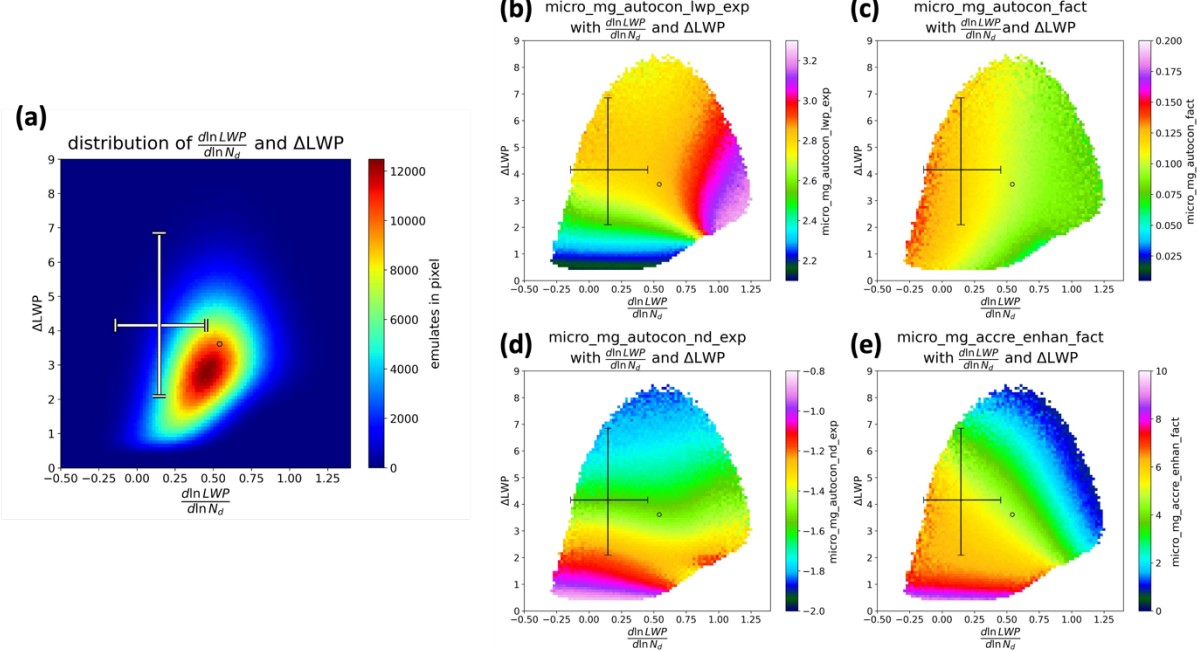

**Figure 14** (a) *The density of emulated distributions and (b-e) how autoconversion-related parameters are distributed. The errorbar represents the 95% confidence interval of the observationally-constrained distribution on each axis and the circle shows the location of the CAM6 default runs. (a) depicts the density of the $10^7$ emulates within the emulator space. (b), (c), and (d) depict the average distributions of* micro_mg_autocon_lwp_exp, micro_mg_autocon_fact, micro_mg_autocon_nd_exp, *and* micro_mg_accre_enhan_fact *within this space. In (b-e) pixels that contained fewer than 50 emulates have been masked.*

## 4 Discussion

Aerosol-cloud adjustments are described in terms of interactions between $N_d$ and LWP, but these processes occur in the context of precipitation and its confounding effects, driven by coalescence scavenging (Figure 1; **Figure 4**). We find that surface observations have utility in constraining global aerosol-cloud adjustments, despite their poor sampling of the global atmosphere (Figure 2). Surface observations from the DOE ARM site at ENA provide a broad suite of cloud and precipitation measurements (Wood et al., 2015) that enable this analysis. Figure 1**Figure 4**

Observed state and covariance metrics examined in this study (Table 4) were within the range produced by the PPE (Figure 6; Figure 8). The regression of LWP on $N_d$ ($\frac{d \ln \text{LWP}}{d \ln N_d}$), which has been used in previous studies to characterize aerosol-cloud adjustments (Bellouin et al., 2020), barely overlapped between the PPE and observations (Figure 8). We share four potential hypotheses to explain this behavior and suggested pathways to evaluate these hypotheses: (i) missing processes in



CAM6, (ii) insufficiently broad PPE parameter priors, (iii) sampling differences between the CAM6 grid cell and the ENA observation footprint, and (iv) observational uncertainty. We briefly discuss each below.

The parameterizations in CAM6 only explicitly address aerosol-cloud adjustments that occur through precipitation suppression. CAM6 does not contain parameterizations that fully treat size-dependent entrainment. Size-dependent entrainment is partially addressed in CAM6 through size-dependent droplet sedimentation (Morrison and Gettelman, 2008). An increase of CCN in a cloud, decreasing the average droplet size, decreases the effect of sedimentation which, in turn, increases entrainment. Through this process there is a partial representation of the processes leading to size-dependent
entrainment in CAM6. These relationships are illustrated succinctly in Figure 1 of Karset et al., 2020.

The lack of a complete parameterization of size-dependent entrainment is common across GCMs (Jing et al., 2019; Karset et al., 2020; Michibata and Takemura, 2015).  One possibility would be to implement size-dependent entrainment in a future version of the PPE, following previous studies, although implementation of this parameterization in a GCM was not found to substantially affect adjustment strength (Karset et al., 2020). A challenge in this approach will be determining if the
structure of the size-dependent entrainment parameterization is reasonable and selecting cases and regimes with sufficient measurements to constrain the size-dependent entrainment process to isolate it from other confounding processes. We stress that to establish the necessity of a size-dependent entrainment parameterization to accurately represent aerosol-cloud adjustments appears to require careful analysis to distinguish between thinning due to size dependent entrainment and non-causal anticorrelation between Nd and LWP driven by precipitation scavenging  (Mahfouz et al., 2024; McCoy et al., 2020a;
Mülmenstädt et al., 2024b).

While the prior distribution of $\frac{d \ln \text{LWP}}{d \ln N_d}$ is mostly more positive than observations, the PPE prior and observations from ENA do overlap. A simple explanation of the occurrence of observed $\frac{d \ln \text{LWP}}{d \ln N_d}$ at the edge of the PPE prior may be that the prior distribution for the CAM6 PPE $N_d$ exponent (micro_mg_autocon_nd_exp, Table 1) – a parameter that governs most of the aerosol-cloud adjustment process in CAM6 (shown in Figure 14c for CAM6 and in other GCMs (Jing et al., 2019)) – may
have been too narrow when the CAM6 PPE was originally designed (Eidhammer et al., 2024). Given the dependence of $\frac{d \ln \text{LWP}}{d \ln N_d}$ on *micro_mg_autocon_fact* and *micro_mg_accre_enhan_fact*, in future iterations of this and other PPEs examining aerosol-cloud adjustments we suggest considering high autoconversion and accretion enhancement factors. This should create more ensemble members with negative $\frac{d \ln \text{LWP}}{d \ln N_d}$. This approach would be the most useful combined with the implementation of a size-dependent entrainment parameterization as discussed above to evaluate the relative importance of these two processes in
producing observed present-day cloud and precipitation behavior.

Another source of disagreement may be the disparity in scale between the CAM6 grid and the ARM sampling footprint. Model output at finer spatial resolutions would allow characterization of the tolerance when comparing GCM grid cell properties to ENA observations. One possibility would be to leverage large eddy simulations (LES) to characterize the relationship between observations at a point to the GCM grid scale. At the time of writing, the LES ARM Symbiotic Simulation




and Observation (LASSO) project (Gustafson et al., 2020) for ENA is in the planning phase and may be useful for future constraint studies. LES in combination with ENA observations would enable further quantification of the impact on adjustments, and more broadly ERFaci, of sub-grid scale processes that are not explicitly parametrized in GCMs. For example, ENA aerosol-cloud-precipitation systems are influenced by varied mesoscale cloud organization (McCoy et al., 2023; Zhou and Bretherton, 2019) and sometimes are buffered against precipitation removal by the presence of small, Aitken mode aerosols

(McCoy et al., 2024). Both of these mechanisms influence the radiative properties and responses to aerosols of the cloud systems but their resulting behaviors are incompletely represented in CAM6 (McCoy et al., 2021, 2023; Zhou et al., 2021).

From an observational perspective, there remains uncertainty related to observation (or lack thereof) of light precipitation and virga. Drizzle and virga conditions – two precipitation regimes for which the disdrometer is inadequately equipped to observe – are prevalent at ENA (Wu et al., 2020). To account for this, future work in this area should utilize remote

sensing retrievals such as Ka-band ARM zenith radar (KAZR) reflectivity (Ghate and Cadeddu, 2019; Wu et al., 2020) to account for these otherwise-missed precipitation events. The reflectivity product available from CAM6 is not adequate to make a comparison to ENA's KAZR. To facilitate this comparison, instrument simulators such as the Earth Column Collaboratory (EMC[2]) (Silber et al., 2022) are required, and may be a promising avenue for future constraint studies motivated by our finding that precipitation played an important role in our constraint of aerosol-cloud adjustments.

In summary, there are several avenues we can take to build on the constraint framework laid out here. However, based on our findings, in a narrow sense we did not find a structural disagreement between ENA observations and the CAM6 model. Critically, although the negative correlation between $N_d$ and LWP is used to support a prevalent thinning of cloud in response to increased aerosol in our assessments of aerosol forcing (Bellouin et al., 2020), we do not find that this is necessarily the case.

**5 Summary**

We present a framework for constraining aerosol-cloud adjustments using mean-state variables and covariances (Table 4). Our framework unites causally-ambiguous present-day observations and a perturbed parameter ensemble (PPE) hosted in the CAM6 global climate model (GCM) to (i) provide constraints on aerosol-cloud adjustments in liquid water path (LWP) as well as (ii) link this constraint to different parameterized processes. Observations from the Eastern North Atlantic

(ENA) were used to constrain global mean aerosol-cloud adjustments. This constraint is the result of selecting model configurations where precipitation rate (P), liquid water path (LWP), droplet number concentration ($N_d$) and their covariance: $\frac{d \ln LWP}{d \ln N_d}$, $\frac{d \ln LWP}{d \ln P}$, and $\frac{d \ln N_d}{d \ln P}$ extracted from the PPE at ENA match their observed equivalents. Response in global-mean LWP to anthropogenic aerosol is constrained to be between 2.08 g/m[2] to 6.87 g/m, a 15% reduction from the prior range in the PPE. Within this constrained emulator space, we see constraint (based intuitively on the shape of the constrained distribution) on 18

out of 45 of the perturbed parameters (Figure 13). Constrained parameters match our *a priori* expectations for processes that are relevant to aerosol cloud adjustments and set cloud and precipitation states. These processes include the autoconversion



parameterization that drives aerosol-cloud adjustments in GCMs (Jing et al., 2019); the accretion parameterization, which is comparable with constraining confounding linkages between LWP, precipitation, and $N_d$; and cloud and convection parameters that are important for setting the mean-state cloudiness and precipitation.

As demonstrated in Figure 14, confounding effects from coalescence scavenging (Gryspeerdt et al., 2019; McCoy et al., 2020a) can operate in conjunction with autoconversion-driven precipitation suppression to reproduce this negative correlation between LWP and $N_d$. We stress that our results do not necessarily rule out size-dependent evaporation and entrainment as an important process in setting aerosol-cloud adjustments, but we do find that this process is not necessary to produce observed present-day behavior and present day observations of clouds and precipitation at ENA are consistent with a
moderate increase in cloud liquid water path in response to anthropogenic aerosol.

**Data Availability**

Data were obtained from the Atmospheric Radiation Measurement (ARM) user facility, a U.S. Department of Energy (DOE) Office of Science user facility managed by the Biological and Environmental Research Program. Data may be downloaded at https://www.arm.gov/data/. The following data citations are provided for each product: Atmospheric Radiation Measurement
(ARM) user facility. 2014. Droplet number concentration (NDROPMFRSR). 2014-06-01 to 2019-10-30, Eastern North Atlantic (ENA) Graciosa Island, Azores, Portugal (C1). Compiled by L. Riihimaki, S. McFarlane, C. Sivaraman and D. Zhang. ARM Data Center. Data set accessed 2024-07-07 at http://dx.doi.org/10.5439/1131339. ; Atmospheric Radiation Measurement (ARM) user facility. 2014. MWR Retrievals with MWRRET Version 2 (MWRRET2TURN). 2014-05-01 to 2024-01-31, Eastern North Atlantic (ENA) Graciosa Island, Azores, Portugal (C1). Compiled by K. Gaustad and D. Zhang. ARM Data
Center. Data set accessed 2024-07-07 at http://dx.doi.org/10.5439/1566156. ; Atmospheric Radiation Measurement (ARM) user facility. 2014. Video Disdrometer VAP (VDISQUANTS). 2014-10-31 to 2024-07-03, Eastern North Atlantic (ENA) Graciosa Island, Azores, Portugal (C1). Compiled by J. Hardin, S. Giangrande, T. Fairless and A. Zhou. ARM Data Center. Data set accessed 2024-07-07 at http://dx.doi.org/10.5439/1592683. CAM6 PPE data (https://doi.org/10.26024/bzne-yf09) is available at https://data.ucar.edu/dataset/cesm2-2-cam6-perturbed-parameter-ensemble-ppe .

**Author Contributions**

AM, DTM, and HM participated in conceptualization and methodology. AM and DTM performed formal analysis. DTM and HM led project administration and funding acquisition. AM and DTM: writing -original draft preparation. AM, DTM, HM, ILM, AG, TE: writing – review and editing. AG, DTM, AM, CS: data curation.



**Competing Interests**

The authors declare that they have no conflict of interest.

**Acknowledgements**

AM, HG, and DTM were supported by the U.S. Department of Energy's Atmospheric System Research Federal Award DE-SC002227 and DTM was supported by U.S. Department of Energy's Established Program to Stimulate Competitive Research DE-SC0024161. ILM was supported by NOAA cooperative agreements NA17OAR4320101 and NA22OAR4320151. CS
was supported by NASA Grant 80NSSC21K2014. We would like to acknowledge the use of computational resources (doi:10.5065/D6RX99HX) at the NCAR-Wyoming Supercomputing Center provided by the National Science Foundation and the State of Wyoming and supported by NCAR's Computational and Information Systems Laboratory. The Pacific Northwest National Laboratory is operated for the U.S. Department of Energy by the Battelle Memorial Institute under contract DE-AC05-76RL01830. TE was supported by the National Aeronautics and Space Administration (grant no. 80NSSC17K0073 and
80NSSC21K1499). The statements, findings, conclusions, and recommendations are those of the author(s) and do not necessarily reflect the views of NOAA or the U.S. Department of Commerce.

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
