# Peer review of "Constraining Aerosol-Cloud Adjustments by Uniting Surface Observations with a Perturbed Parameter Ensemble"

_EGUsphere, 2024_

## Referee Comment (RC1)

**Review of "Constraining Aerosol-Cloud Adjustments by Uniting Surface Observations with a Perturbed Parameter Ensemble" by Mikkelsen et al.**

This paper aims to reduce the uncertainty in aerosol-cloud adjustments by constraining a PPE of a GCM with surface observations of cloud properties from the East North Atlantic (ENA). The surface observations include the variables LWP, precipitation and $N_d$. The PPE is produced from the atmospheric component (CAM6) of CESM2. Gaussian process emulators are created for the median, covariances and PI-PD change in modeled cloud properties. Strong correlations are found between the median state and the covariances and some of the PPE parameters e.g. $N_d$ and subgrid velocity. The emulated properties are then constrained by the observed cloud properties at the ENA surface station, and the impact of the constrained parameters on the change in global mean PI-PD LWP is shown. The constrained parameters include processes that are relevant for ACI in GCMs.

The study looks to constrain aerosol-cloud adjustments in a novel way though using surface observations and emulated model output. It's a nice piece of analysis and well written. I recommend the minor corrections below before publication.

Comments

Figure 2 caption: specify that Figure 2a is from the model.

P8, L196: What was the approach in Eidhammer et al. 2024 designed to sample the uncertainty in and how does that correspond to the parameters that are likely important for ACI?

P9, L205-210: Some more information on the model set is needed to improve this section. How long are the simulations for PD and PI? Do the simulations have all anthropogenic and emissions set to PD and PI respectively, or is it just the anthropogenic aerosol emissions that change between the two simulations? Why was the year 2000 used for PD when the observations are from a different year? How useful is a constraint trend on years with different emissions? What is used for prescribing SSTs, sea ice and land cover?

P9, L209: What model levels are nudged?

P9, L213: It the model output analysed in the gridbox for the ENA surface site is in or is it interpolated in some way?

P10, L218: Does aerosol impact convective precipitation in this model?

P20, L360: Could you remind the reader what criteria is used to rule out an emulate again here, I had to go back to the methods to look it up. Or it might be clearer to reference the constraint process in its own subsection in the methods?

Figure 12 and discussion: In Section 3.1 it is noted how the observable covariation between Nd and LWP in PD may not be predictive of PI-PD LWP changes driven by Nd, and the differences in the predictability of delta Nd for delta LWP between the ENA and globally. These limitations are not really mentioned around Figure 12 or in the discussion. I.e. can you really be confident that constraint of model parameters based on PD observations from one site are representative of PI-PD global changes?

P24, L415: Typo – figure 14 is repeated a few times.

---

## Referee Comment (RC2)

**Review of** *Constraining Aerosol-Cloud Adjustments by Uniting Surface Observations with a Perturbed Parameter Ensemble*, Mikkelsen et al.

**Summary:** This manuscript presents a use case of the CAM6 perturbed parameter ensemble (PPE), deployed to constrain aerosol-cloud adjustments in conjunction with surface observations from an atmospheric observatory. A set of previously established parameters are perturbed within the model, and a series of emulators are then developed to sample the space defined by the 45 parameters. Leveraging cloud and precipitation observations, the authors are able to constrain the range of global delta LWP from preindustrial to present day, relevant to understanding the effective radiative forcing of the climate. Importantly, the range of the global aerosol-cloud adjustments is entirely positive, implying a cooling over the historical period. Minor revisions before publication are suggested.

**Revisions:**
1. Section 1 (95-105) and Section 2.1 (125-130) both contain discussions about the utility of surface observations to constrain aerosol-cloud adjustments. Suggest re-arranging and/or merging these comments and keeping 2.1 focused on the features of the ENA observatory.
2. While the paper focuses largely on the LWP response to aerosol perturbations, no mention is made of the cloud fraction (CF) adjustment. Typically, SW ERFaci is decomposed into the Twomey effect, LWP, and CF adjustments. The emphasis on cloud water changes is clear throughout the paper, but the CF adjustment might merit a mention in the Introduction to fully address the scope of what "aerosol-cloud adjustments" implies to readers.
3. Section 2.1.2 mentions that the observations of surface rain rate may miss those in the "drizzle domain," and Figure 6 shows that the ENA observations approach the upper range of PPE values for mean rain rate. How might a more accurate precipitation observation (that captures drizzle) impact this result, and can you be certain that differences can be attributed to sampling (grid cell mean vs station)? If the precipitation observations were out of the range of the PPE, how would that change your analysis? Given the importance placed on mean-state precipitation as a constraint (line 370), this may bear further discussion.
4. How does the 15% reduction in spread of global delta LWP compare to the results in Song et al. (2024)? Would you expect the constraint found here to be weaker due to the use of surface observations as a constraint? The use of an identical PPE could be cause for assessing the constraint found here.
5. How relevant is a constraint on the spread of global delta LWP when the prior range is entirely positive, given that there is a well-documented tendency for GCMs to predict uniformly positive LWP responses to aerosol perturbations, which disagrees with conclusions from many observational studies? There is relatively little acknowledgement of this in the introduction and conclusion, but it merits consideration.
6. Can the findings in Figure 13 be utilized by CAM6 developers for parameter tuning the next version of CAM? And can any of your findings inform future PPE studies - are there parameter ranges that should be widened or narrowed? Are there parameters that

should be omitted due to showing little influence on ACI and related processes? Some of this is briefly discussed in line 510, but could be expanded to widen the implications of this study.

a) Line 469: (2) and (6 → (2) and (6)
b) Line 483: Figure1Figure4 may be a typo
c) Line 416: Figure4 is repeated. Typo
d) Line 548: 6.87 g/m to g/m2
e) Figure 3: bottom right panel may be duplicated or cropped incorrectly
f) Some figure labels are unclear whether they refer to ENA or global. Suggest adding subscripts to denote for clarity.

---

## Author Response (AR1)

Reviewer questions are in *orange*, author responses are in blue.

**Referee #1**

*Figure 2 caption: specify that Figure 2a is from the model.*

Thanks for the catch – this has been updated in the manuscript. Corrections are in the Figure 2 caption.

*P8, L196: What was the approach in Eidhammer et al. 2024 designed to sample the uncertainty in and how does that correspond to the parameters that are likely important for ACI?*

The approach in Eidhammer et al., 2024, was designed to investigate uncertainties in subgrid scale processes relating to cloud microphysics, turbulence, convection and aerosols. These processes are important to aerosol-cloud interactions through their intertwined effects on cloud formation, aerosol activation, aerosol emissions, cloud formation, and precipitation formation. This has been expanded upon in lines 214-221.

*P9, L205-210: Some more information on the model set is needed to improve this section. How long are the simulations for PD and PI? Do the simulations have all anthropogenic and emissions set to PD and PI respectively, or is it just the anthropogenic aerosol emissions that change between the two simulations? Why was the year 2000 used for PD when the observations are from a different year? How useful is a constraint trend on years with different emissions? What is used for prescribing SSTs, sea ice and land cover?*

Thanks for pointing this out. 1995-2005 anthropogenic emissions averages have been used, set as such to be consistent with Eidhammer et al., 2024 and AeroCom standards. While we acknowledge that there are differences in aerosol emissions between this average and the modeled period, we don't think they are significant enough that the constrain would be affected. SST/sea ice are fixed to 1995-2005 averages. Land cover is run through the coupled CESM land model. Clarifications to this point are added in lines 230-233.

*P9, L209: What model levels are nudged?*

The model is nudged at all levels. This has been clarified in line 235.

*P9, L213: It the model output analysed in the gridbox for the ENA surface site is in or is it interpolated in some way?*

The model output is analyzed in the gridbox for the ENA surface site. This has been clarified in lines 239 and 240.

*P10, L218: Does aerosol impact convective precipitation in this model?*

While we did not examine convective precipitation specifically in this study, the convection scheme in this model (ZM) is not directly impacted by drop number or activation. Results from Eidhammer et al., 2024, (see figure 8) which has the same PPE setup, shows this succinctly. A clarifying point has been added in line 246.

*P20, L360: Could you remind the reader what criteria is used to rule out an emulate again here, I had to go back to the methods to look it up. Or it might be clearer to reference the constraint process in its own subsection in the methods?*

Thanks for pointing this out- we've added a quick refresher (and internal reference to the relevant section) in the part of the paper describing the constraint results. See lines 395-397.

*Figure 12 and discussion: In Section 3.1 it is noted how the observable covariation between Nd and LWP in PD may not be predictive of PI-PD LWP changes driven by Nd, and the differences in the predictability of delta Nd for delta LWP between the ENA and globally. These limitations are not really mentioned around Figure 12 or in the discussion. I.e. can you really be confident that constraint of model parameters based on PD observations from one site are representative of PI-PD global changes?*

Thanks for bringing this up, it warrants further discussion. One way to interpret the result from Figure 12 is that the PD observations from this one site are ultimately not very strongly predictive of ΔLWP. After constraining to the 6 observations from ENA, the spread of $\Delta LWP_{gl}$ is only reduced by ~15% - This can potentially be attributed to the inherent limitations in constraining to a single surface observatory as opposed to broader climatology. While ENA (and other similar surface observatories) provide a unique and useful venue for observing fine-scale processes, it should not be expected that ENA alone will be able to tightly constrain $\Delta LWP_{gl}$. For clues as to the path forward, it is useful to compare the results of this study to Song et al., 2024, which constrains $\Delta LWP_{gl}$ in the same PPE to satellite measurements of globally-averaged LWP, $N_d$, and upwelling top-of-atmosphere shortwave radiation. When comparing the constrained $\Delta LWP_{gl}$ ranges between these two studies, while neither constraint is especially strong on its own, each constraint rules out different extremes in $\Delta LWP_{gl}$. This work shows a larger minimum constrained $\Delta LWP_{gl}$ (2.08 g/m$^2$); Song et al., 2024, shows a smaller maximum constrained $\Delta LWP_{gl}$ (4.33 g/m$^2$). While this should not be considered a strict scientific comparison, it suggests that a more robust constraint can be found by constraining on the fine-scale (e.g., surface-observed precipitation) and the broader climate-scale (e.g. satellite-observed average global upwelling shortwave radiation) simultaneously. This is an idea we will be exploring in future

work. This additional discussion has been added to the manuscript in the Discussions section, see lines 531-550. We've also added a brief clarifying point to this at lines 520 and 521.

*P24, L415: Typo – figure 14 is repeated a few times.*

Fixed, thank you for catching that.

**Referee #2**

*Section 1 (95-105) and Section 2.1 (125-130) both contain discussions about the utility of surface observations to constrain aerosol-cloud adjustments. Suggest re-arranging and/or merging these comments and keeping 2.1 focused on the features of the ENA observatory.*

Thanks for the feedback, we have re-arranged and combined these sections for clarity and conciseness. This can be seen in lines 106-112 and in the shortened introduction for Section 2.1.

*While the paper focuses largely on the LWP response to aerosol perturbations, no mention is made of the cloud fraction (CF) adjustment. Typically, SW ERFaci is decomposed into the Twomey effect, LWP , and CF adjustments. The emphasis on cloud water changes is clear throughout the paper, but the CF adjustment might merit a mention in the Introduction to fully address the scope of what "aerosol-cloud adjustments" implies to readers.*

Thank you for the feedback, we have added some additional clarification to this point in the Introduction. See lines 40, 43, and 44.

*Section 2.1.2 mentions that the observations of surface rain rate may miss those in the "drizzle domain," and Figure 6 shows that the ENA observations approach the upper range of PPE values for mean rain rate. How might a more accurate precipitation observation (that captures drizzle) impact this result, and can you be certain that differences can be attributed to sampling (grid cell mean vs station)? If the precipitation observations were out of the range of the PPE, how would that change your analysis? Given the importance placed on mean-state precipitation as a constraint (line 370), this may bear further discussion.*

We are not certain whether the difference in precipitation between models and observations stems from sampling vs. instrument capability. If we had a more accurate precipitation observation, though, we could at least eliminate one of these issues, allowing us to home in on process disagreements. If this caused us to get observations that were outside of the range of the PPE, we would need to expand our PPE to a broader range of parameter values. However, we would expect a more accurate observation to bring the mean precipitation down towards the bulk of the PPE distribution, because it would see a larger number of light rain observations. Because we cannot adequately separate these uncertainties at this time, however, the observation stands as-is. Additional clarification to these points has been added to the Discussion section of the manuscript, lines 601-605.

*How does the 15% reduction in spread of global delta LWP compare to the results in Song et al. (2024)? Would you expect the constraint found here to be weaker due to the use of surface observations as a constraint? The use of an identical PPE could be cause for assessing the constraint found here.*

Thanks for the feedback on this. Generally, this result is in line with results of Song et al., 2024. Both constraints on ΔLWP are relatively weak and constrain CAM6 towards stronger adjustments. We would also agree with the idea that the $\Delta LWP_{gl}$ constraint in Song et al., 2024, is stronger because they are using global observations rather than point-source observations. While we attempted to control for a larger regime with the state variables (e.g. median LWP) at ENA, ultimately controlling for observations from a single point on the planet, even one well-correlated with $\Delta LWP_{gl}$, may not be strong enough to significantly constrain uncertainty in this area. One interesting contrast between our $\Delta LWP_{gl}$ constraints is that, with the constrained distributions taken together, we rule out different extremes. In this work, we see a larger minimum constrained value ($2.08$ g/m$^2$); in Song et al., 2024, a smaller maximum constrained value is found ($4.33$ g/m$^2$). While we would not put forth this exact comparison as a precise scientific result, we posit this is representative of the benefits that unifying these styles of analyses can have: by combining the benefits from constraining on the fine-scale (e.g., surface-observed precipitation) and the broader climate-scale (e.g. satellite-observed average global upwelling shortwave radiation) simultaneously, a more robust constraint can be found. Further discussion on this point has been added in the beginning of the Discussions section of the manuscript, lines 531-550.

*How relevant is a constraint on the spread of global delta LWP when the prior range is entirely positive, given that there is a well-documented tendency for GCMs to predict uniformly positive LWP responses to aerosol perturbations, which disagrees with conclusions from many observational studies? There is relatively little acknowledgement of this in the introduction and conclusion, but it merits consideration.*

Results from many present-day observational studies regarding LWP responses to aerosol perturbations tend to occur in highly specific environments from which it is difficult to extrapolate PD-PI changes, and causality is not so easily observed (e.g. Christensen et al., 2017). Furthermore, it has been shown that PD variability in aerosol-cloud interactions that can create these sorts of "negative relationships" between $N_d$ and LWP can be replicated in models with positive adjustments (Mülmenstädt et al., 2024). So, following Mülmenstädt et al., 2024, and the results from Figure 14, it is not considered problematic that this model uniformly produces positive $\Delta LWP_{gl}$ because this result is ultimately not in contradiction of these observational studies. To help better convey this point, additional clarification has been added to the end of the Discussions section, lines 610-612.

*Can the findings in Figure 13 be utilized by CAM6 developers for parameter tuning the next version of CAM?*

Yes- we think that if used in concert with Figure 10 from Eidhammer et al., 2024 (which is a similar collection of constrained histograms) CAM6 developers may be able to make some specific inferences about parameters that can be tuned towards better representation of aerosol-cloud interactions. We specify "in concert with Eidhammer et al., 2024" because, going off the thought process described in our response to *"How does the 15% reduction in spread of global delta LWP compare to the results in Song et al. (2024)?"* we think it is prudent for model developers to utilize responses from fine- and broad-scale constraints simultaneously. Tuning parameters where the constraints seem to agree strongly (e.g., clubb_C2rt) may be a good place to start. Additional commentary in service of this point has been added to the Discussions section in lines 544-550.

*And can any of your findings inform future PPE studies - are there parameter ranges that should be widened or narrowed?*

From Figure 13, we would suggest that the prescribed parameter range for zmconv_capelmt should be limited to a smaller range in future PPEs for CAM6. We have added this recommendation to the Results section surrounding Figure 13. Following the discussion to the previous question, we cannot confidently recommend changes to any of these ranges otherwise. Added this recommendation in lines 450-452.

*Are there parameters that should be omitted due to showing little influence on ACI and related processes? Some of this is briefly discussed in line 510, but could be expanded to widen the implications of this study.*

We think that would depend on the experiment. If we were to rerun a PPE for this experimental setup, then yes- we would remove all parameters that had a flat distribution, tighten the distributions of the remaining parameters towards the constraint, and rerun the ensemble. But for future work, as discussed in prior responses, we are interested in simultaneously constraining off fine- and broad- scale observations for a more holistic and robust climate-wide aci constraint. It's possible that some of these other parameters, which didn't show a strong response in this particular regime, may be important when constraining across these broader climate levels. So, in this case, we would be hesitant in removing them altogether. This discussion is also in 544-550.